# Compositional generalization through abstract representations in human and artificial neural networks

**Takuya Ito**[*]
Yale University
taku.ito1@gmail.com

**Tim Klinger**
IBM Research AI
tklinger@us.ibm.com

**Douglas H. Schultz**
University of Nebraska-Lincoln
dhschultz@unl.edu

**John D. Murray**
Yale University
john.murray@yale.edu

**Michael W. Cole**
Rutgers University
michael.cole@rutgers.edu

**Mattia Rigotti**
IBM Research AI
mr2666@columbia.edu

## Abstract

Humans have a remarkable ability to rapidly generalize to new tasks that is difficult to reproduce in artificial learning systems. Compositionality has been proposed as a key mechanism supporting generalization in humans, but evidence of its neural implementation and impact on behavior is still scarce. Here we study the computational properties associated with compositional generalization in both humans and artificial neural networks (ANNs) on a highly compositional task. First, we identified behavioral signatures of compositional generalization in humans, along with their neural correlates using whole-cortex functional magnetic resonance imaging (fMRI) data. Next, we designed pretraining paradigms aided by a procedure we term *primitives pretraining* to endow compositional task elements into ANNs. We found that ANNs with this prior knowledge had greater correspondence with human behavior and neural compositional signatures. Importantly, primitives pretraining induced abstract internal representations, excellent zero-shot generalization, and sample-efficient learning. Moreover, it gave rise to a hierarchy of abstract representations that matched human fMRI data, where sensory rule abstractions emerged in early sensory areas, and motor rule abstractions emerged in later motor areas. Our findings give empirical support to the role of compositional generalization in human behavior, implicate abstract representations as its neural implementation, and illustrate that these representations can be embedded into ANNs by designing simple and efficient pretraining procedures.

## 1 Introduction

Humans can efficiently transfer prior knowledge to novel contexts, an ability commonly referred to as transfer learning. One proposed mechanism underlying transfer learning is compositional generalization (or compositional transfer) – the ability to systematically recompose learned concepts into novel concepts (e.g., "red" and "apple" can be combined to form the concept of a "red apple") [5, 9, 20]. Indeed, it has been suggested that an algorithmic implementation of compositional generalization is one of the key missing ingredients that ANN models need in order to achieve human-like learning and reasoning capabilities [30, 28]. Therefore, quantifying how compositional generalization is manifested in human behavior and investigating its underlying implementation in biological brains is a natural first step to harness and deploy it in machine learning models.

---

[*]Work done as an intern at IBM Research AI

36th Conference on Neural Information Processing Systems (NeurIPS 2022).

Recent studies that investigated compositionality in machine learning have typically relied on architectures comprised of specialized modules. For instance, disentangled representation learning separates the independent factors underlying the structure of the input data into disjoint components of the feature vector [17, 18, 35, 16]. Program synthesis methods achieve state-of-the-art performance on systematic generalization [20] through model architectures built by combining specialized neural and symbolic program modules interacting to search over a space of valid production rules [29, 37, 46].

Complementing these studies, *abstract representations* have been recently proposed as vector representations that reconcile compositional generalization with distributed neural codes [2]. In particular, *parallel abstract representations* – representations with a high Parallelism Score as previously defined [2] – support out-of-context generalization by encoding changes in individual variables as a linear shift in the representations. This notion of abstraction implies that these representations are compositionally additive; novel compositions are encoded as the vector sum of distinct abstract representations. This is similar to how word2vec embeddings solve relational analogy tasks [34, 31] and generalizes disentangled representations by allowing for arbitrary affine transformations of disentangled codes. Crucially, this type of representation is operationally defined in a way that can be quantified in neuroimaging data by computing the Parallelism Score metric defined in [2]. In other words, parallel abstract representations are a computationally promising candidate as neural substrate implementing compositional generalization, and are also measurable in the human brain by computing the Parallelism Score across fMRI voxels during neuroimaging experiments.

This work is motivated by the working hypothesis that parallel abstract representations support compositional generalization. Accordingly, we first characterized the behavioral signatures of compositional generalization in a task that systematically varied rule conditions across 64 contexts, showing that humans generalize better to tasks with greater similarity structure to previous tasks. We then analyze fMRI imaging data showing that parallel abstract representations are distributed across the entire cortex in a content-specific way during the execution of the compositional task. This supports our working hypothesis that parallel abstract representations may implement compositional generalization in humans. To test this hypothesis in ANNs, we designed a pretraining paradigm for ANNs to emulate humans' prior knowledge about the compositional task elements, finding that ANNs pretrained in this way exhibit 1) more abstract representations, 2) excellent generalization performance, and 3) sample-efficient learning. This finding demonstrated that the degree of abstraction (induced through pretraining) directly impacted zero-shot compositional generalization performance in ANNs. Finally, we find that the layerwise organization of abstract representations in pretrained ANNs recapitulates the content-specific distribution across the human sensory-to-motor cortical hierarchy. Together, these findings provide empirical evidence for the role of abstract representations in supporting compositional generalization in humans and ANNs.

## 1.1 Related work

Several recent studies in neuroscience have applied analytic tools to identify the neural basis of rapid generalization in biological neural networks. Such studies employed various measures – cross-condition generalization [2, 40, 8, 4], state-space projections of task-related compositional codes [51, 43, 25], and Parallelism Score [2] – to quantify the generalizability and abstraction of representations. Prior work in neuroscience has primarily evaluated compositionality in limited context settings (e.g., up to 10 contexts), or without manipulating different types of features (e.g., higher-order vs. sensory/motor features). Moreover, these neuroscience studies used simple task paradigms due to limitations in either the model organism (rodents and monkeys are unable to perform complex tasks [2]) or to isolate specific types of abstraction in humans (e.g., logical abstractions [40]). Here we significantly expand on prior work by using a 64-context compositional task that systematically varies different types of task features (e.g., sensory, motor, and logical rules) to evaluate content-specific abstractions across the entire brain and multilayer ANNs. This work also complements related work in compositional generalization in machine learning [28, 46, 20, 29, 45, 50, 19]. However, those studies primarily focused on building models that improve on current compositional generalization benchmarks on arbitrarily complex compositional tasks, such as SCAN [28], COG [50], or GQA [19]. Importantly, these studies did not directly benchmark ANN behavior (or representations) against human behavioral and neural data, making a direct comparison difficult. Here we leveraged a non-trivial 64-context compositional paradigm to investigate the representations that facilitate compositional generalization in both humans and ANNs.

## 2 Methods

### 2.1 The Concrete Permuted Rule Operations (C-PRO) task paradigm

We used the C-PRO paradigm (Fig. 1a) during fMRI acquisition and ANN model training. Briefly, the C-PRO paradigm permutes specific task rules from three rule domains (logical decision, sensory semantic, and motor response) to generate dozens of novel task contexts. This creates a context-rich dataset in the task context domain. The sensory rule indicates which stimulus feature should be attended to. The logic rule specifies a Boolean operation to be implemented on the stimulus feature set. The motor rule specifies a specific motor action (i.e., a button press with a specific finger). One of 256 possible unique stimulus combinations could be presented with each task context. Visual dimensions included either horizontal or vertical bars with either blue or red coloring. Auditory dimensions included continuous (constant) or beeping high or low pitched tones.

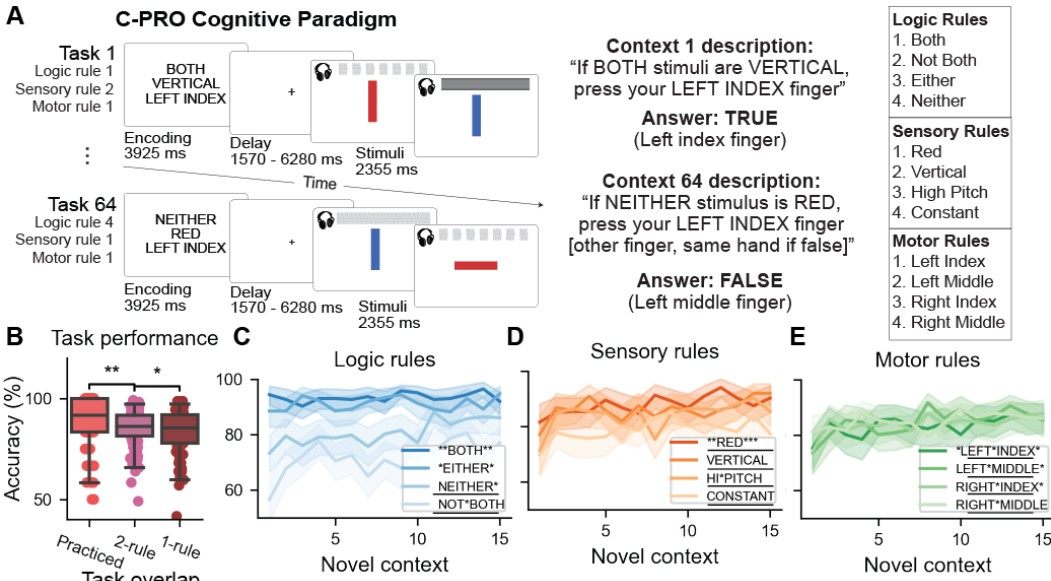

Figure 1: a) The C-PRO paradigm permutes 12 rules belonging to three different rule domains – logical, sensory, and motor gating – to generate up to 64 unique contexts. b) Human performance on novel task contexts was significantly lower than on practiced contexts (participants were trained on four practice contexts prior to the test session). Moreover, subjects performed novel task contexts with more rule overlap with practiced contexts at a higher accuracy. c-e) Task performance as a function of task trials for each rule (novel contexts only). Consistent with compositional generalization, participants had a significant increase in task performance in 10/12 rules, even though each rule was used in a novel context. Shaded area around line plots (c-e) reflects the 95% confidence interval.

Each rule domain (logic, sensory, and motor) consisted of 4 specific rules (Fig. 1a). A task context is comprised of one rule from each domain, for a total of 64 possible task contexts (4 logic x 4 sensory x 4 motor). Subjects were trained on 4/64 "practiced" contexts prior to the fMRI session. The 4 practiced contexts were selected such that all 12 rules were equally practiced. Subjects' mean performance across all trials was 84% (median=86%; chance=25%). See Appendix A.2 for details[1].

### 2.2 The geometry of abstract neural representations

Behavioral signatures of compositional generalization can be investigated by measuring behavioral performance as a function of task composition and prior learning. Neural signatures of generalization can be identified using analysis methods that characterize the geometry of neural activations during task generalization. In particular, prior work proposed the Parallelism Score (PS) [2] as a measure to evaluate the consistency of task variable representation across contexts. Intuitively, PS identifies a coding axis (i.e., the parallel displacement vector) across task contexts that aids generalization.

---

[1]The fMRI dataset is publicly available here: https://openneuro.org/datasets/ds003701

We posit that representations with high PS (the specific type of abstract representation we investigate) support compositional generalization in human behavior. We illustrate here how PS is reflected in the geometry of neural representations with respect to the rule domains of the C-PRO task. Let us consider a set of C-PRO contexts with logic rules BOTH or EITHER, and sensory rules with values RED or VERTICAL (Fig. 2). High PS in the logic rule domain indicates that the difference in activation vectors between contexts with BOTH and EITHER rules is the same when paired with either the RED or VERTICAL sensory rules. Thus, a change from BOTH to EITHER results in the same parallel change irrespective of the sensory rule (Fig. 2c). In contrast to unstructured high-dimensional representations (Fig. 2a, and see [41, 12]), this would afford high generalization, since the effect of changing the logic rule in either sensory rules automatically transfers to the other sensory rule.

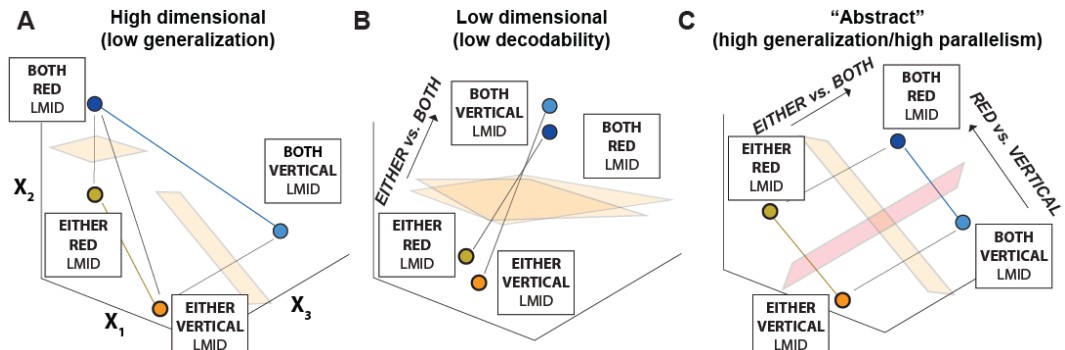

Figure 2: Hypothetical geometric configurations of neural activation space for "BOTH vs. EITHER" and "RED vs. VERTICAL" rule contrasts. a) High-dimensional representations of task activations lead to low PS (in addition to low generalizability across conditions) of rules. b) Low-dimensional representations lead to overall low decodability, but some generalizability (across limited features). c) Parallel Abstract Representation of the neural activations leads to high generalizability.

## 2.3 Parallelism score

We generalize the definition of PS by [2] to tasks where variables can assume an arbitrary number of values (as opposed to being binary) and applied it to human fMRI and internal ANN activations. PS is defined as the cosine angle of the coding directions of the same rules in different contexts in the neural activation space (e.g., voxels or neurons within a brain region) (Fig. 3a-c). A cosine angle close to 1 indicates coding directions that are highly parallel, despite differences in context. We compute the coding angle for a specific rule dichotomy (e.g., the coding direction "BOTH" vs. "EITHER") by identifying all pairs of task contexts that had exactly the same secondary (sensory) and tertiary (motor) rules. For each pair, we subtracted the fMRI voxel activation vectors associated with each context to obtain the vector that represented that coding direction (see Fig. 3a). We did this for all other pairs in that coding direction. Defining $v_i$ as this coding vector for the $i$th pair, we computed the PS score for one dichotomy as $PS_k = \frac{1}{16} \sum_{i \neq j}^{16} cos(v_i, v_j))$, since there are 16 possible pairs for each rule dichotomy within the C-PRO task. To obtain the PS for a specific rule domain (e.g., logic, sensory, or motor rules), $PS_k$ is computed for every coding direction, then averaged (e.g., for logic PS, the average of "BOTH" versus "EITHER", "BOTH" versus "NEITHER", etc.).

Statistical testing was performed using a non-parametric procedure, where we shuffled labels within each rule domain 1000 times and re-calculated PS to produce a null distribution. We corrected for multiple comparisons (across brain regions) using non-parametric family-wise error correction [36].

## 2.4 ANN construction and training

The primary ANN architecture had two hidden layers (128 units each) and an output layer that was comprised of four units that corresponded to each motor response (Fig. 4; see Appendix section A.7 for additional details). Training used a cross-entropy loss function and the Adam optimizer [27]. The ANN transformed a 28-element input vector into a 4-element response vector with the equation $Y = f_{ReLU}(X_h W_h + b_h)$. Weights and biases were initialized from a uniform distribution $U(-\sqrt{1/k}, \sqrt{1/k})$, where $k$ is the number of input features from the previous layer.

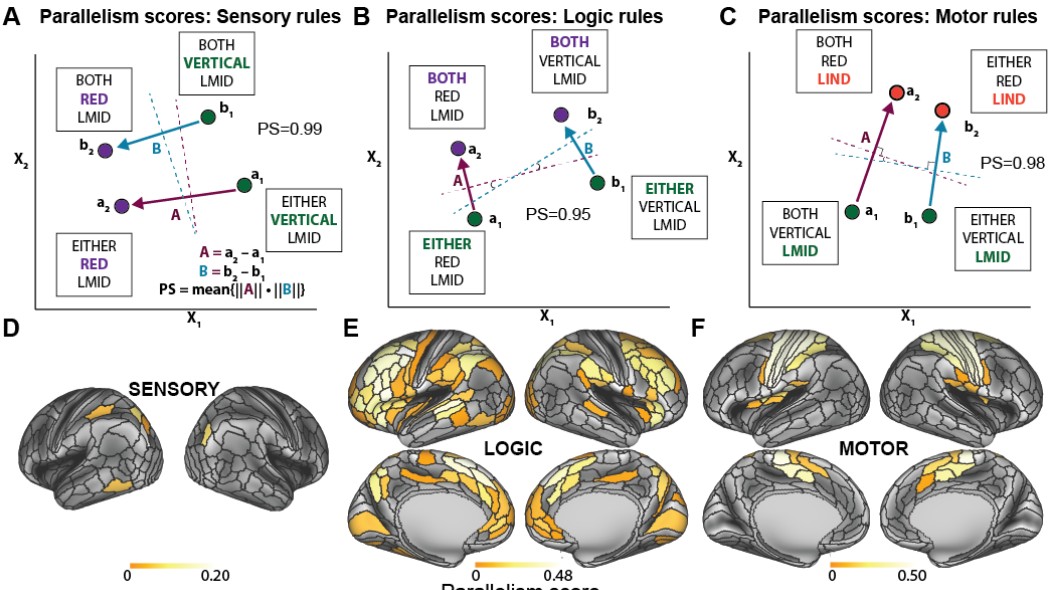

Figure 3: a-c) 2-D schematic visualization of PS estimation for the a) sensory, b) logic, and c) motor rule domains for a specific rule pair (e.g., RED vs. VERTICAL). Intuitively, PS captures the geometry of the neural activation space (ANN or fMRI data) by measuring the cosine angle between two linear decoders trained to distinguish two rule conditions in different task contexts. d-f) PS was calculated for each rule domain for every brain region [13]. PS was highest in association areas for logic rules, dorsal attention network regions for sensory rules, and somatomotor network for motor rules.

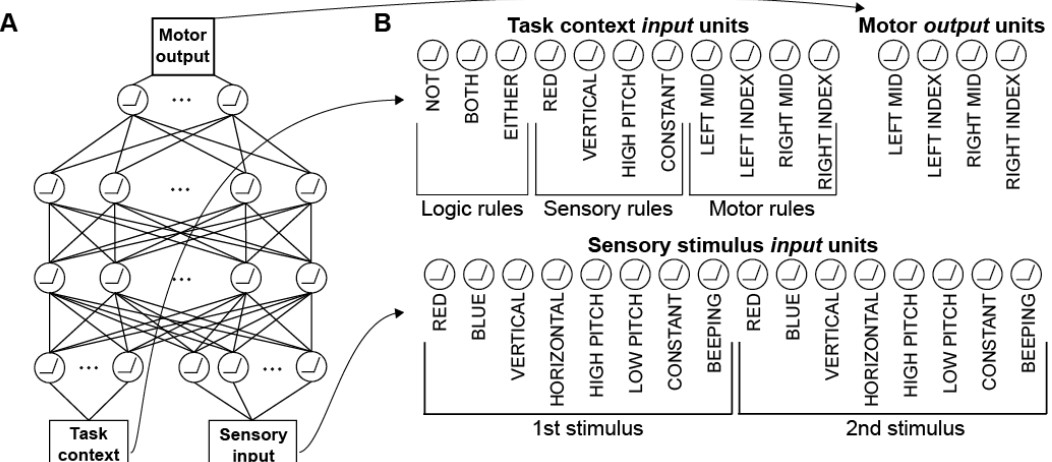

Figure 4: a) ANN architecture. We employed a simple architecture with two hidden layers (128 units each) with rectified linear units. b) The ANN's input and output units. Task input units were modeled as one-hot vectors for each rule across all rule domains. Sensory stimulus input units were modeled as categorical one-hot vectors for each stimulus type. Separate vectors were modeled for first and second stimulus presentations. Motor outputs had a unit corresponding to each motor response. Noise was injected with input features when calculating PS to reduce the orthogonality of input features.

Training on the C-PRO task was performed in a sequential learning paradigm. To mimic the human experiment, an arbitrary set of four practiced contexts was initially selected for training. (This was randomly selected across different ANN initializations.) Then, novel task contexts were incrementally added into the set of training contexts.

# 3 Results

## 3.1 Behavioral signatures of rapid compositional generalization in humans

We evaluated human behavioral compositional generalization by assessing performance on novel contexts in the C-PRO paradigm. Since adult humans have decades of prior knowledge, subjects were able to compositionally generalize to novel task contexts without any training (novel accuracy=84.17%, chance=25%, Wilcoxon signed-rank p<0.0001). However, subjects performed the four practiced contexts better than novel contexts (practiced=87.67%, novel=84.17%; p=0.003). We next assessed how performance on novel contexts changed as a function of shared rule structure to the practiced contexts. Consistent with compositional transfer of previously learned rules, performance on novel task contexts improved as a function of similarity to the practiced contexts (accuracy, 2-rule overlap=84.86%; 1-rule overlap=83.48%; practiced vs. 2-rule overlap, p=0.008; 2-rule vs. 1-rule overlap, p=0.03; Fig. 1b). Though our findings are consistent with compositional transfer, we found that rapid transfer to novel contexts is more difficult. However, we found that increased exposure to specific rules improved performance on subsequent novel contexts using that same rule (all except for the "Both" and "Either" rules, likely due to ceiling effects, FDR-corrected p<0.05; Fig. 1c-e). This suggests that even though performance in novel contexts is worse than practiced contexts, subjects can improve rule transfer with increased practice (or pretraining).

## 3.2 Spatial and content-specific topography of abstract representations in human cortex

We extended prior work to identify abstract representations using PS across the entire human cortex [2]. We calculated PS for each rule domain separately (Fig. 3a-c) using the vertices/voxels within each parcel (i.e., brain region) as activation vectors. We found topographic differences of sensory, logic, and motor rule abstractions tiled across human cortex (Fig. 3d-f). Specifically, we found that statistically significant sensory rule abstractions were primarily identified in higher order visual areas and the dorsal attention network (i.e., brain areas involved in the top-down selection of visual stimuli) (PS of significant regions=0.15; family-wise error (FWE)-corrected p<0.05; Fig. 3d). Logic rule abstractions were more widely distributed, but primarily observed in frontoparietal areas (PS of significant regions=0.22; FWE-corrected p<0.05; Fig. 3e). Motor rule abstractions were primarily localized to somatomotor cortex (PS of significant regions=0.29; FWE-corrected p<0.05; Fig. 3f). Notably, regions with abstract representations form a subset of regions of those that contain rule information using standard decoding methods (SFig. 1). This ensures that high PS is accompanied by highly decodable representations (i.e., high dimensionality).

## 3.3 Embedding prior knowledge into ANNs with simple pretraining tasks

Human behavioral data suggested improved compositional generalization with increased task rule exposure, in addition to the years of "pretraining" from ordinary development (i.e., at least 18+ years). Thus, we sought to evaluate whether embedding prior knowledge of rules could improve compositional generalization in ANNs, while simultaneously investigating how prior knowledge impacts the geometry of ANNs' internal task representations. Given that the C-PRO task was specifically designed as a compositional task that conjoined three task rules, we created pretraining paradigms designed to teach ANNs basic rule knowledge (Fig. 5; see Appendix for full description).

We constructed a simple feedforward ANN with two hidden layers (Appendix A.7; Fig. 4). This made it easier to investigate the effects of pretraining on internal representations, rather than architectural choices. We designed two pretraining paradigms: Primitives (1-rule) and Simple task (2-rule) pretraining. Primitives pretraining trained on 1-rule tasks that focused explicitly on learning the semantics of primitive rule features (Fig. 5a-c). This included distinguishing sensory stimuli, learning motor response mappings (e.g., "left index" rule would lead to a left index response), and abstract logical relations, which involved learning the boolean relations amongst logic rules. Simple task pretraining focused on learning 2-rule conjunctions (i.e., a sensory and motor rule pairing / logical and sensory rule pairing) (Fig. 5d-e). Importantly, these pretraining paradigms focused on learning primitive 1- or 2-rule associations that were significantly simpler than the full C-PRO task (3-rule combination).

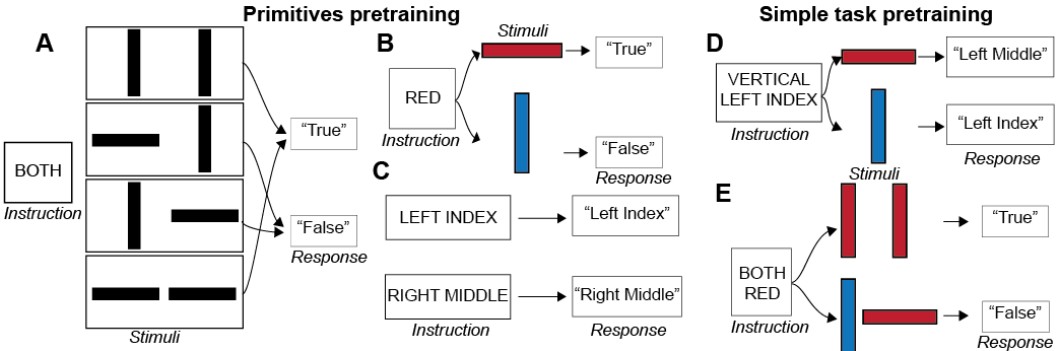

Figure 5: a) The logic rule primitives task involved teaching boolean relations among different logical operations. For example, when presented with the "BOTH" rule, the task was to distinguish two identical ("True") versus two different ("False") stimuli (i.e., same vs. different). b) Sensory rules involved mapping sensory rules onto stimulus features. c) Motor rules involved mapping motor rules onto motor output units. d-e) Simple task pretraining (2-rule tasks) was designed to teach the model how to perform simple (d) sensorimotor mappings and (e) logical-sensory gatings.

## 3.4 Pretraining induces abstractions, zero-shot performance, and sample efficiency

We measured the PS in ANNs trained with different pretraining routines: Vanilla (no pretraining), Primitives pretrained, Simple task pretrained, and Combined (Primitives + Simple task pretrained). Each pretraining condition was trained until ANNs achieved 99% accuracy (see A.8 for details). PS was calculated for each rule domain using the ANN's hidden layer activations (see A.9). Pretrained ANNs had higher PS than the Vanilla ANN (Primitives vs. Vanilla, t(37)=5.26, p=1e-05; Simple task vs. Vanilla, t(37)=8.46, p=1e-11; Combined vs. Vanilla, t(37)=3.03, p=0.003) (Fig. 6a). Moreover, PS increased from Primitives to Simple task pretraining (t(37)=3.91, p=0.0002), though no significant increase in PS was observed in Combined vs. Simple task pretraining.

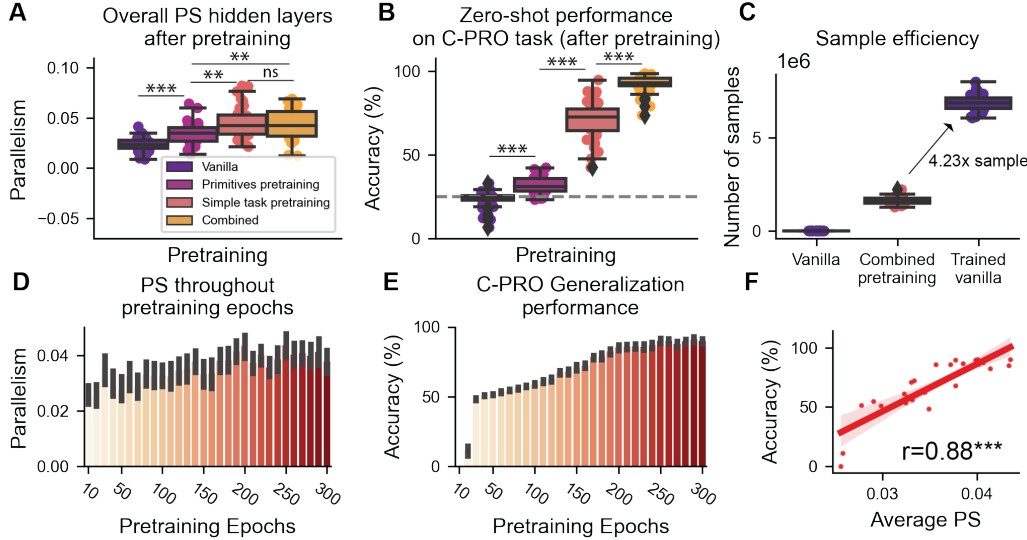

Figure 6: a) PS of hidden units averaged across all rule domains using noisy inputs. b) Zero-shot learning of all 64 C-PRO contexts. c) Sample efficiency of models (Combined and trained vanilla model were performance-matched). Total samples, including pretraining samples (if applicable). d) We computed the average PS of the ANN hidden layers throughout Combined pretraining. (Note that PS values are low due to input features containing large amounts of noise.) e) Zero-shot performance to the unseen C-PRO trials throughout pretraining. c) The generalization performance and the PS scores were highly correlated with each other throughout pretraining, indicating a close link between abstract representations and improved task generalization.

We next evaluated the zero-shot performance on the full C-PRO task after pretraining (Fig. 6b). As expected, the Vanilla ANN performed near chance (acc=23.25%, chance=25%, one-sided t(38)=-2.17, p=0.98). Primitives pretraining marginally improved zero-shot performance (acc=31.51%, t(38)=8.09, p<1e-9). While Simple task pretraining exhibited significant improvement over Primitives pretrained models (acc=70.57%, Simple task vs. Primitives, t(37)=19.84, p<1e-31), Combined pretraining had excellent zero-shot performance on the entire C-PRO task (acc=92.15%, Combined vs. Simple task pretraining, t(37)=10.85, p<1e-16).

We next sought to assess the impact of pretraining on learning/sample efficiency. We therefore trained a Vanilla network (no pretraining) on 60/64 C-PRO contexts to match the zero-shot performance of the Combined pretraining model (i.e., at least 90% accuracy on the 60 context training set). We found that on the remaining test set (4/64 C-PRO contexts), the Vanilla trained model achieved 96.02% generalization performance, but required up to 4.23x training samples to match the performance of the Combined model (Fig. 6c). Critically, the 4.23x more training samples included all possible samples (pretraining and C-PRO samples). Thus, pretraining afforded both zero-shot generalization and sample efficient learning.

Finally, we measured PS and generalization performance throughout Combined pretraining. We found that PS and zero-shot performance increased with pretraining (Fig. 6d,e), and were highly correlated (r=0.88; p<10e-9; Fig. 6f). This illustrated that the abstract representations learned during pretraining directly facilitated zero-shot generalization, and is consistent with prior work demonstrating that the dimensionality of hidden representations is altered throughout training [39].

## 3.5 ANN pretraining leads to human-like compositional generalization

We evaluated the learning and generalization dynamics of ANNs with and without pretraining, after training ANNs on 4 of the full C-PRO contexts. This matched the human experiment, since humans were exposed to 4 "practice" contexts prior to performing the remaining 60 novel contexts (Fig. 1b). (ANN training was stopped after achieving 90% performance on the 4 practiced contexts.) We found overall poor generalization on novel task contexts in the Vanilla model (accuracy, practiced=94.37%, novel=28.79%; p<0.0001; Fig. 7a). This suggested that ANNs with no prior knowledge cannot compositionally generalize. We subsequently compared generalization performance on ANNs after pretraining. We found that with Primitives pretraining, generalization performance significantly improved (57.97%; Fig. 7b). We observed additional improvements with Simple task pretraining (86.79%; Fig. 7c), achieving human-like generalization performance (Fig. 7d).

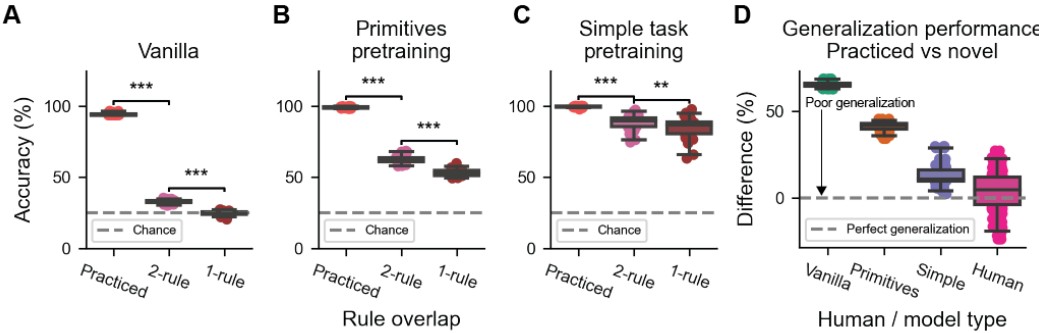

Figure 7: a-c) We trained the ANN architecture on 4/64 C-PRO task contexts with (a) a Vanilla ANN with randomly initialized weights, (b) an ANN after Primitives pretraining, and (c) an ANN after Simple task pretraining. Training on the 4 practiced contexts was stopped after the model achieved 90% accuracy on those contexts. As in our empirical human data (Fig. 1b), the ANN performed best on contexts in which it had previously seen (Practiced), followed by contexts with two overlapping (2-rule) and one overlapping (1-rule) rules. d) Generalization performance of ANNs with Primitives and Simple task pretraining was most similar to human generalization performance in novel contexts. (Lower values indicate better generalization.)

We next incrementally trained all ANN models on novel contexts, by adding one novel context into the training set at a time. We tested generalization performance on the held-out (test set) contexts until ANNs were trained on 63/64 contexts (SFig. 2a). Generalization performance on novel contexts

was significantly higher in ANNs with either pretraining routine (SFig. 2b). This was despite the fact that all ANNs had the same stopping criteria (i.e., 90% accuracy on the C-PRO training set). We ran an additional experiment where each of the ANNs were shown a fixed number of C-PRO task samples during training, replicating our core finding (SFig. 3). These findings suggest that the inductive biases formed during pretraining significantly improve downstream generalization performance.

### 3.6 Pretraining ANNs facilitates sample-efficient learning throughout novel task learning

We sought to evaluate how pretraining impacted sample efficiency. We found that pretrained ANNs became more sample efficient as the training set expanded, even after accounting for total number of (pretraining and C-PRO) samples (SFig. 2b). We quantified the generalization performance to sample efficiency ratio as the generalization inefficiency, finding that after learning only 7 C-PRO contexts, vanilla ANNs generalized worse than pretrained ANNs (SFig. 2c). These findings support the notion that pretraining can simultaneously improve compositional generalization and sample efficiency.

### 3.7 Convergent hierarchy of abstract representations in humans and ANNs

Analysis of human fMRI data revealed that content-specific abstraction was spatially heterogeneous across cortex. Recent neuroscience work has identified hierarchical gradients that organize along a sensory-to-motor output axis in both resting-state [32] and multi-task fMRI data [21]. We therefore sought to quantify PS across this sensory-to-motor hierarchy in fMRI data, and compare it to PS changes in the feedforward hierarchy (i.e., layer-depth) in ANNs. We focused our analyses on the Combined pretrained model (which incorporates both Primitives and Simple task pretraining) due to its excellent zero-shot generalization (Fig. 6b). In addition, we extended our model to include three hidden layers to make it easier to compare PS of different hidden-layer depths to the three cortical systems of interest: sensory, association, and motor systems (Fig. 8a).

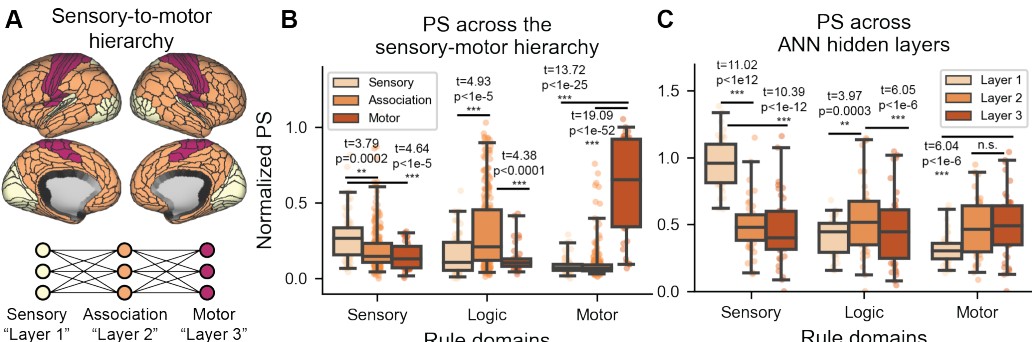

Figure 8: a) A discretized sensory-to-motor hierarchy (see SFig. 6 for discretization details). b) We computed the normalized PS (i.e., the PS of each brain region normalized by the maximum PS across all regions) for each rule domain across the discretized cortical systems. c) Same analysis as in b), but using the PS found in each ANN hidden layer.

We measured the PS for each rule domain for sensory, association, and motor systems. Sensory rule PS was highest in the sensory system, logic rule PS was highest in association systems, and motor rule PS was highest in the motor system (Fig. 8b). To observe whether similar hierarchical PS organization emerged in ANNs, we used the Combined pretrained model with three hidden layers, and plotted PS as a function of ANN depth. Since our ANN transformed sensory inputs into motor outputs, we analogized each ANN layer to the sensory, association, and motor cortical systems (Fig. 8a). We found a similar pattern in the ANN: sensory PS peaked in the first hidden layer; logic PS peaked in the second hidden layer; and motor PS peaked in the last two hidden layers (Fig. 8d). We corroborated these findings using a continuous sensory-motor hierarchical gradient map (without discretization) (SFig. 5-6). (In addition, see SFig. 7-9 for PS scores for all possible rule dichotomies in human fMRI data and ANN activations.) These findings suggest that abstraction emerges as a function of rule-dependent specialization and hierarchical organization.

# 4 Discussion, Limitations, Conclusions

We provide empirical support for the role of compositionality in human generalization, and implicate abstract representations as its neural implementation. In classic ANNs, which are known to perform poorly during systematic generalization [20, 6], we found that computationally cheap pretraining paradigms embedded abstract representations that led to human-like generalization performance and sample efficient learning. When mapping abstract representations across cortex and ANN layers, we found converging patterns of rule-specific abstractions from early sensory areas/layers to late motor areas/layers across human and ANN hierarchies. These results reveal the hierarchical organization of content-specific abstractions in the human brain and ANNs, while revealing the impact of these abstractions for compositional generalization in models.

Our pretraining approach directly leverages knowledge of task structure to design pretraining routines that embed task biases into ANNs. Despite the sample efficiency of this approach, this pretraining approach requires the initial overhead of designing paradigms useful for downstream learning. A related approach that similarly requires prior knowledge of task structure is "representational backpropagation" – a regularization approach that aims to produce an idealized hidden representation [26]. However, there are other inductive bias approaches that do not require prior task knowledge. One approach constrains ANNs to produce abstract task representations by initializing ANN weights from a low-norm distribution [8]. However, initializing ANN weights in this regime is computationally costly. Another approach is to initialize networks with built-in modular structures to facilitate the re-use of network modules across tasks [33, 44]. However, exactly how such networks disentangle representations has not yet been explored. Nevertheless, all these approaches are complementary to each other. It will be important for future work to assess how these approaches may synergistically interact to optimize for sample-efficient generalization in multi-task settings.

Though we provide comprehensive evidence of the role of abstraction in compositional generalization, there are several limitations in the present study that future research can explore. We found that the spatial topography of abstract representations was highly content-dependent. However, analyses were limited to cross-context manipulations of limited rule types (sensory, logic, and motor gating), without addressing the organization of other task components (e.g., reward or stimuli). Thus, future studies can explore how brains and ANNs represent the abstraction of other task components. Second, though we were able to explore cross-context generalization across 64 contexts – significantly more than previous empirical studies in neuroscience – cross-context analysis was limited to a single task type (i.e., the C-PRO paradigm). It will be critical to see the organization of abstraction in multi-task settings that go beyond 64 contexts. Finally, our ANN modeling approach revealed the computational benefits of pretraining. It will be important for future work to benchmark sample efficiency and generalization performance against other training paradigms (e.g., in continual learning and/or meta-learning settings; [15, 49]).

In conclusion, we characterized a convergent hierarchical organization of abstract representations across the human cortex and in ANNs using a 64-context paradigm, and provided insight into the impact of abstract representations on generalization performance. Overall, we found that simple pretraining tasks efficiently embed abstract representations into ANNs, leading to improved systematic generalization similar to human behavior. These findings provide a human-centric benchmark from which to understand compositional generalization in ANNs, paving the way for greater interpretability of compositionality in ANNs. Importantly, investigating compositionality through a human-centric framework (e.g., by benchmarking ANNs against humans in the same task) creates a concrete target for interpreting the strengths and limitations of compositionality in ANNs. We hope these findings inspire further investigations into the comparison of compositionality in humans and ANNs.

## Acknowledgments and Disclosure of Funding

T.I. acknowledges a research internship sponsored by IBM Research and a Swartz Foundation postdoctoral fellowship from Yale University. This project was supported in part by the US National Institutes of Health, under awards K99-R00 MH096901 and R01 MH109520 (M.W.C.). The content is solely the responsibility of the authors and does not necessarily represent the official views of any of the funding agencies.

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
