# A  Supplementary Methods

## A.1  Participants

The following description is quoted with citation from a previous study that used the same dataset [23]. The dataset was publicly published under a CC0 license, and is publicly available (https://openneuro.org/datasets/ds003701). All participant data on the public repository has been de-identified.

Data were collected from 106 human participants across two different sessions (a behavioral and an imaging session). Technical error during MRI acquisition resulted in removing six participants from the study. Four additional participants were removed from the study because they did not complete the behavior-only session. fMRI analysis was performed on the remaining 96 participants (54 females). All participants gave informed consent according to the protocol approved by the Rutgers University Institutional Review Board. The average age of the participants that were included for analysis was 22.06, with a standard deviation of 3.84.

## A.2  C-PRO task paradigm – additional details

The C-PRO cognitive paradigm permutes specific task rules from three different rule domains (logical decision, sensory semantic, and motor response) to generate dozens of novel and unique task contexts. Visual stimuli included either horizontally or vertically oriented bars with either blue or red coloring. Simultaneously presented auditory stimuli included continuous (constant) or non-continuous (non-constant, i.e., "beeping") tones presented at high (3000Hz) or low (300Hz) frequencies. A given task context could be presented with 256 unique stimulus combinations. This is because a given task context was presented with two sequentially presented audiovisual stimuli, where each audiovisual stimulus varied in four dimensions: color (red/blue), orientation (vertical/horizontal), pitch (high/low), continuity (continuous/beeping). This led to $2^8 = 256$ possible stimulus combinations. The paradigm was presented using E-Prime software version 2.0.10.353 [48].

Each rule domain (logic, sensory, and motor) consisted of four specific rules, while each task context was a combination of one rule from each rule domain. A total of 64 unique task contexts (4 logic rules x 4 sensory rules x 4 motor rules) were possible, and each unique task set was presented twice for a total of 128 task miniblocks. This meant that there were $256 * 64 = 16384$ unique trials (i.e., context-stimulus) combinations. Identical task contexts were not presented in consecutive blocks. Each task miniblock included three trials, each consisting of two sequentially presented instances of simultaneous audiovisual stimuli. A task block began with a 3925 ms encoding screen (5 TRs), followed by a jittered delay ranging from 1570 ms to 6280 ms (2-8 TRs; randomly selected). Following the jittered delay, three trials were presented for 2355 ms (3 TRs), each with an inter-trial interval of 1570 ms (2 TRs). A second jittered delay followed the third trial, lasting 7850 ms to 12560 ms (10-16 TRs; randomly selected). A task block lasted a total of 28260 ms (36 TRs). Subjects were trained on four of the 64 task contexts for 30 minutes prior to the fMRI session. The four practiced rule sets were selected such that all 12 rules were equally practiced. There were 16 such groups of four task sets possible, and the task sets chosen to be practiced were counterbalanced across subjects. Subjects' mean performance across all trials performed in the scanner was 84% (median=86%) with a standard deviation of 9% (min=51%; max=96%). All subjects performed statistically above chance (25%).

## A.3  Analysis of human task performance data

The corresponding results described in this section can be found in Fig. 1.

We calculated the average accuracy for each task miniblock (comprising three task trials). Note that each task miniblock had the same task context (three-rule combination) for all three trials. This resulted in 128 task accuracy scores for every subject. Task rule contexts were then sorted into three categories separately for every subject: practiced, 2-rule overlap, and 1-rule overlap. Practiced task contexts were defined as the four task contexts that were used to train participants on the C-PRO task outside of the MRI scanner. 2-rule overlap tasks were task contexts that had at least 2 of the same rules overlapping with the previously seen practiced tasks, and 1-rule overlap tasks were tasks with only a 1-rule overlap with practiced tasks. Note that there were no 0-rule overlap tasks, since subjects were trained on every rule prior to the test session. Moreover, there were no rule overlaps across practiced task contexts. Finally, every participant was provided with a randomly selected set of practiced task contexts. Behavioral accuracy was computed for every task context group for every subject (Fig. 1b).

We next evaluated the successive miniblock performance of each rule presented in a novel task context during the fMRI scanning session (see Fig. 1c-d). This would capture a participant's ability to use a previously seen rule in a novel context as a function of the number of times it used the rule previously. The performance of an individual rule (e.g., "BOTH") was calculated separately per participant as a function of each novel context seen. Thus, performance of each rule was calculated for exactly 15 miniblocks (each rule was presented 16 times, including the practiced miniblock). For each rule, we then fit a linear regression model to assess whether performance (dependent variable) could be calculated as function of miniblock presentation (independent variable) with a

positive coefficient (i.e., increasing slope). A significant positive increase of performance (increasing positive performance) was tested for significance by assessing the p-value of the beta coefficient ($p<0.05$ threshold). This captured compositional learning – re-using a task rule (despite use in a novel context) indicated that participants were learning to use previously learned rules in out-of-set novel contexts.

## A.4 fMRI acquisition

The following fMRI acquisition details are taken from a previous study that used the same data set [23].

Whole-brain multiband echo-planar imaging (EPI) acquisitions were collected with a 32-channel head coil on a 3T Siemens Trio MRI scanner with TR=785 ms, TE=34.8 ms, flip angle=55°, Bandwidth 1924/Hz/Px, in-plane FoV read=208 mm, 72 slices, 2.0 mm isotropic voxels, with a multiband acceleration factor of 8. Whole-brain high-resolution T1-weighted and T2-weighted anatomical scans were also collected with 0.8 mm isotropic voxels. Spin echo field maps were collected in both the anterior to posterior direction and the posterior to anterior direction in accordance with the Human Connectome Project preprocessing pipeline [14]. A resting-state scan was collected for 14 minutes (1070 TRs), prior to the task scans. Eight task scans were subsequently collected, each spanning 7 minutes and 36 seconds (581 TRs). Each of the eight task runs (in addition to all other MRI data) were collected consecutively with short breaks in between (subjects did not leave the scanner).

## A.5 fMRI preprocessing

The following details are quoted with citation from a previous study that used the same preprocessing scheme [23].

Resting-state and task-state fMRI data were minimally preprocessed using the publicly available Human Connectome Project minimal preprocessing pipeline version 3.5.0. This pipeline included anatomical reconstruction and segmentation, EPI reconstruction, segmentation, spatial normalization to standard template, intensity normalization, and motion correction. After minimal preprocessing, additional custom preprocessing was conducted on CIFTI 64k grayordinate standard space for vertex-wise analyses using a surface based atlas [14]. This included removal of the first five frames of each run, de-meaning and de-trending the time series, and performing nuisance regression on the minimally preprocessed data [3]. We removed motion parameters and physiological noise during nuisance regression. This included six motion parameters, their derivatives, and the quadratics of those parameters (24 motion regressors in total). We applied aCompCor on the physiological time series extracted from the white matter and ventricle voxels (5 components each extracted volumetrically) [1]. We additionally included the derivatives of each component time series, and the quadratics of the original and derivative time series (40 physiological noise regressors in total). This combination of motion and physiological noise regressors totaled 64 nuisance parameters, and is a variant of previously benchmarked nuisance regression models [3].

## A.6 fMRI activation estimation

We performed a within-subject task GLM on the vertex-wise fMRI time series to estimate task rule-related activations on the CIFTI grayordinate space. To extract task activations for each task block, we performed a beta series regression on every task miniblock [42]. Specifically, we fit an independent regressor to every encoding period (3925ms, 5 TRs), resulting in 128 task regressors in total. Fitting regressors on the encoding period was done primarily to isolate rule representations rather than the actual trial (stimulus-response) period. Each regressor was a boxcar function that was a vector of 0s, except for the specified encoding period. This boxcar function was then convolved with the SPM canonical hemodynamic response function [10]. A single activation estimate (beta coefficient) was extracted for every encoding block at every surface vertex.

## A.7 ANN construction and batch training

The primary ANN architecture was comprised of two hidden layers, each with 128 units. The output layer was comprised of four units that corresponded to each motor response. The ANN transformed the trial input vector into a 4-element response vector with the equation $Y = f_{ReLU}(X_h W_h + b_h)$, where $Y$ corresponds to the output vector, $W_h$ is the weight matrix from the last hidden layer to the output layer, $X_h$ is the activation vector of the last hidden layer, and $b_h$ is the bias vector. The hidden unit activation vectors were defined as $X_i = f_{ReLU}(X_{i-1} W_{i-1} + b_{i-1})$, where $X_i$ is the activation vector for layer $i$. Weights and biases were initialized from a uniform distribution $U(-\sqrt{1/k}, \sqrt{1/k})$, where $k$ is the number of input features from the previous layer.

The ANN was optimized by minimizing the cross entropy between the outputs and the correct target output (a one-hot vector). Optimization was performed using Adam with a learning rate of 0.001 [27]. During training, we used dropout (with probability 0.2 from input to hidden, and 0.5 within hidden layers).

Training on the C-PRO task was performed in a sequential learning paradigm. Initially, an arbitrary set of four practiced contexts were selected with the constraint that no set of four practiced contexts had any overlapping

rules. (This was randomly selected across different ANN initializations). Then, novel task contexts were incrementally added (by 1) into the set of training contexts, and performance (and ANN analysis) was performed after the addition of each task context. Training stopped once all 64 task contexts had been fully trained on. We used batch training. Each batch contained a single task context with all possible stimulus (256) combinations. Thus, each batch contained 256 trials in total.

To stop training we set criteria for two different experiments. The first experiment required that performance on each task context (each batch) in the training set achieved a baseline performance accuracy or better (90%). Once this criterion was satisfied, a novel task context would be added into the training. The second experiment kept fix the number of batches/gradient steps each task context took prior to adding a new task context. This was set to 200 gradient steps per task context.

## A.8 ANN pretraining description

There were two pretraining paradigms: Primitives pretraining and Simple task pretraining.

Primitives pretraining focused on training individual rules within each rule domain, enabling ANNs to learn individual rule Primitives (e.g., the notion of what "RED" is). This is consistent with how humans enter the full C-PRO experiment – most participants already know what the primitives "RED", "LEFT MIDDLE", or "BOTH" refer to. While C-PRO task contexts activated 3 rules out of 12 possible task rules, Primitives tasks only activated 1/12 rules. Primitives pretraining was performed for each of the rule domains separately.

Motor Primitives pretraining involved making motor responses given a motor rule. This involved activating one motor rule in the input vector at a time. If the "LEFT MIDDLE" rule was activated, a LEFT MIDDLE output response would be expected. In addition, if the "NOT" unit was activated in conjunction with the "LEFT MIDDLE" rule, then the "LEFT INDEX" output response would be expected, which is the analogous rule instruction that participants received prior to performing the C-PRO task.

Sensory Primitives pretraining involved making True/False statements on whether a specific sensory stimulus feature was presented. This involved activating one sensory rule and one sensory stimulus feature at a time. For example, for the sensory rule "RED", either a RED/BLUE sensory stimulus would activate. During pretraining, we included two additional output units that corresponded to True/False units. If a "RED" stimulus was presented in conjunction with "RED" sensory rule, the output should be "True"; otherwise, "False". In addition, we also presented sensory rules with the "NOT" negation. In other words, if the rule was "NOT RED" and the sensory stimulus presented was RED, then the network should produce "False".

Logic Primitives pretraining involved learning the abstract logical relations between different logic rules. The logic rules "BOTH" and "EITHER" could be equivalently interpreted as "AND" and "OR" logic rules, respectively. The "NOT BOTH" and "NEITHER" rules were analogous to the negations of those rules. This was operationalized by presenting two stimuli from a specific feature domain (e.g., color). Using stimuli from the color feature domain as an example, for "BOTH" Primitive training, if RED-then-RED stimuli were presented, this would result in a "True" boolean. In contrast, if RED-then-BLUE stimuli were presented, then this would result in a "False" boolean. This intuition was derived from conditional logic (where "BOTH" is equivalent to the concept of "SAME"), where the statement $x == x$ is True. In contrast, "EITHER" (i.e., "OR") was coded as "True" for any combination features of a color stimuli. The negations "BOTH" and "EITHER" resulted in the negation of the produced boolean (i.e., "NEITHER" produced "False" for all stimulus pairs). The most critical point, however, is that the responses for all logic rules were distinct among each other, and described equivalent logical relations between rules. Previous theoretical work in cognitive science suggests that symbolic computation emerges by learning the relational representations between symbolic operations [38]. This Logic Primitives pretraining approach captures the relations amongst symbolic operations.

Simple task pretraining involved combining two-rule task context, rather than the full three-rule task context in the C-PRO task. We designed two variants of simple tasks: a sensorimotor task that combined a sensory and motor rule, and a logical-sensory task that combined a logic and sensory rule. The sensorimotor task activated a sensory rule, motor rule, and one sensory stimulus unit. The response output was a motor response. For example, if the rules were "RED" and "LEFT MIDDLE", and the stimulus was "RED", ANNs were taught to respond with the Left Middle response unit. If the stimulus was anything other than "RED", then ANNs were taught to respond with the Left Index response unit. The sensorimotor task was used to teach ANNs simple mappings between sensory input and motor responses in the simplest possible paradigm.

The logical-sensory task was designed to teach ANNs logical inferences over sensory stimuli in the simplest possible manner. Like the sensorimotor task, the logical-sensory task included two rules: a logic and sensory rule. However, unlike the sensorimotor task, it activated two stimulus units from the same feature domain (e.g., the color domain). For example, for the logic rule was "BOTH" and sensory rule "RED", if the first stimulus was red and second stimulus red, the ANN would be taught to respond with the True output unit. If any of the stimuli were not red (e.g., blue), then the ANN would be taught to respond with the False output unit.

Primitives pretraining was always performed prior to Simple task pretraining, except for SFig. 4, which investigated the effect of reversing the order of pretraining tasks. Pretraining procedures were blocked together, such that all conditions within the Primitives pretraining paradigm (i.e., Logic, Sensory, and Motor primitives pretraining) were trained until all three tasks achieved 99% accuracy. Simple task pretraining was subsequently performed until both Logical-Sensory and Sensory-Motor tasks were performed at 99% accuracy. This ordering is consistent with prior work, suggesting that ANNs are more sample efficient when transitioning from easier to more difficult tasks [47]. Conditions within each pretraining protocol were interleaved [11], ensuring that catastrophic forgetting was not an issue, as is common in continual learning paradigms [7]. We also performed a simple control experiment demonstrating that when pretraining was reversed in the Combined condition (i.e., Simple task pretraining followed by Primitives pretraining), generalization performance was reduced to chance (SFig. 4). This suggests that the ordering of pretraining paradigms is crucial for generalization performance and sample efficiency, which future work should explore.

After pretraining, the additional True/False output units were lesioned from the network.

## A.9  ANN analysis

Analysis of ANNs was carried out in a similar manner to how empirical fMRI data and behavior was analyzed. ANN analysis was performed to infer how the structure of their internal representations was associated with task generalization performance and sample efficiency. Task generalization performance was calculated as the performance on novel contexts (i.e., novel recombinations of task rules). This was independent of whether or not ANNs had learned/seen individual rules previously. To estimate task sample efficiency, we calculated the number of trial samples required to achieve a baseline task accuracy percentage (on contexts in the training set). Note that for a fixed number of trial samples, the number of gradient steps were the same (batch sizes were always the same). Generalization inefficiency was measured as the ratio of the number of samples trained on (normalized between 0 and 1) and the generalization performance on novel contexts (normalized to 0 to 1 + a fixed constant).

PS in ANNs was calculated separately from the training procedure. Since we were only interested in the PS of rule representations, only input units associated with the task rules were activated, while stimulus inputs were set to 0. This ensured that the hidden activations were not contaminated by stimulus-related activations when calculating PS in the hidden layers. Otherwise, PS in ANNs was calculated in a similar manner to the empirical fMRI data, where the spatial features (dimensions) were the units within a given hidden layer (like voxels within a brain parcel).

In addition, Supplementary Figures 7, 8, and 9 illustrate that pairwise PS scores for all pairs of rule dichotomies in both ANNs and human fMRI data.

## A.10  Computing resources

fMRI analyses were carried out on a local server with 24 cores and 320GB of RAM. ANN training, while not required, was performed on an NVIDIA P100 GPU. A single ANN initialization can be successfully trained on a CPU in under 2 minutes.

# B  Supplementary Figures

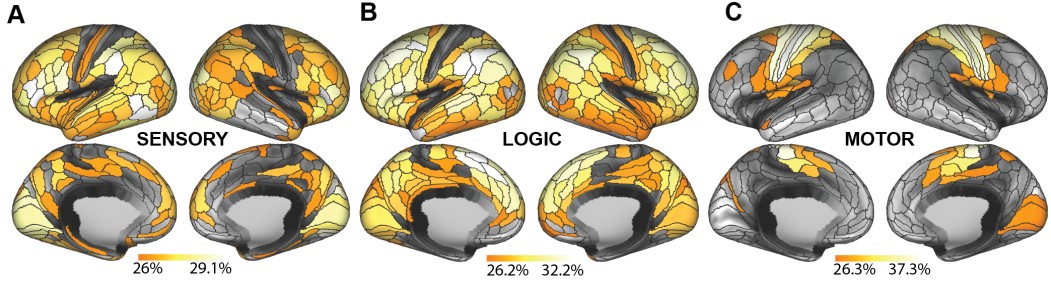

Standard distance-based deocding

Supplementary Figure 1: Standard decoding of multivariate activation patterns reveals more distributed patterns of decodability than identified with PS. These findings suggest that of the regions that contain task information, only a subset of these regions contain abstract representations. a) Sensory, b) logic, and c) motor rule decoding at the group level (n=96). Decoding was performed using a distance-based classifier (Pearson correlation), and significance was assessed using a binomial test against chance (25%). Significance was assessed using multiple comparisons-corrected threshold (False Discovery Rate) of p<0.05.

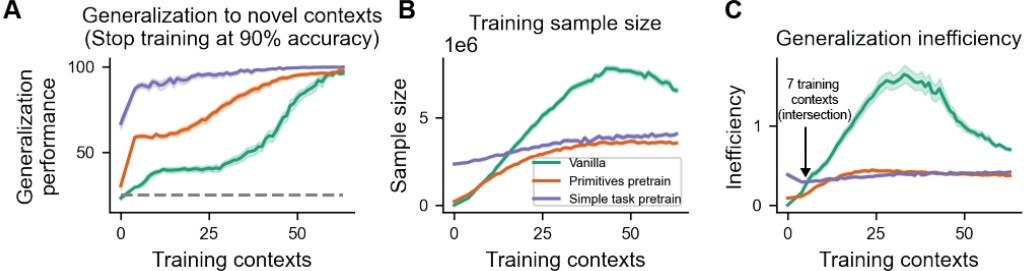

Supplementary Figure 2: a) Generalization performance on novel contexts after training on n/64 contexts (x-axis). b) Number of total trials/samples shown to each model type. c) The generalization inefficiency of each model. Generalization inefficiency was measured as the ratio of the number of samples shown (normalized between 0 and 1) and the performance on novel tasks (normalized to 0 to 1 + a fixed constant). See SFig. 3 for model performance using a fixed number of samples per trained context.

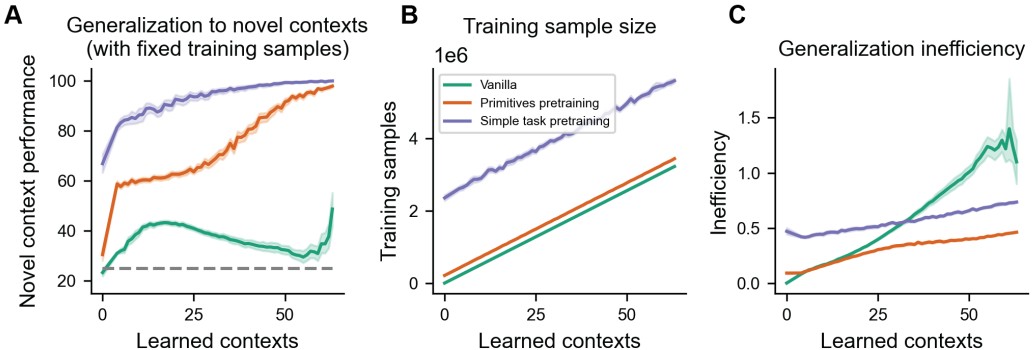

Supplementary Figure 3: Novel context performance as function of learned contexts in the C-PRO task in the ANN model. Similar to SFig. 2, but we trained the model using a fixed number of samples per learned context (200 epochs per context) and evaluated generalization performance. a) We systematically trained each type of ANN on a range from 0-63 of the C-PRO task contexts, and assessed its generalization on the remaining (novel) contexts. We trained the model on a fixed number of samples per learned context. Learned contexts were sequentially introduced, and novel context performance was assessed on the remaining (excluded from training) contexts. b) The number of total trials/samples shown to each model type. Since the number of samples shown to the model for each context was fixed, training samples linearly increased as a function of learned contexts. c) As in SFig. 2c, we measured the generalization inefficiency of each model. Thus, while the Vanilla model was initially more efficient, pretrained models quickly learned more efficiently as evidenced by better generalization accuracies with fewer samples.

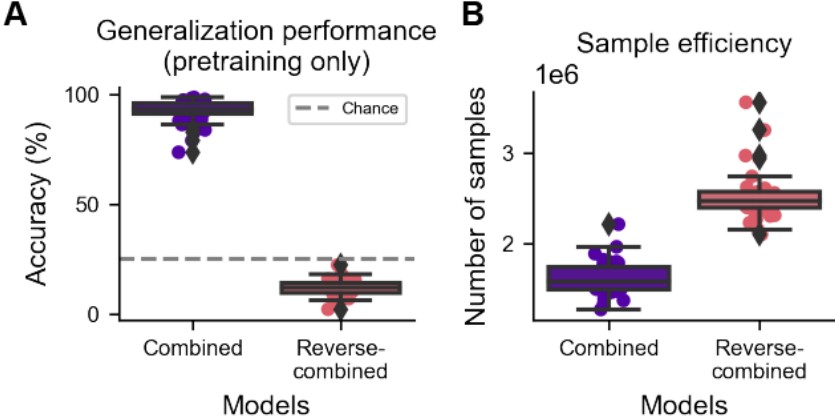

Supplementary Figure 4: Order of pretraining paradigms drastically affects generalization performance and sample efficiency. In the Combined pretraining regimen, we first implemented Primitives pretraining followed by Simple task pretraining. However, prior curriculum learning research suggests that the order of task learning can significantly impact generalization performance [47]. Thus, we compared the Combined pretraining protocol to a Reverse-combined protocol, where Simple task pretraining was performed prior to Primitives pretraining. a) We found that generalization performance on the unseen C-PRO tasks was significantly impaired in the Reverse-combined condition. b) We also measured the sample efficiency of only the pretraining trials. We found that Reverse-combined pretraining required more samples than Combined pretraining, despite the stopping criterion for each pretraining task remaining the same. (ANNs were required to perform either Primitives pretraining or Simple task pretraining with at least 99% accuracy prior to moving on to the next pretraining task.) This suggests that ordering of pretraining tasks can significantly impact the learning and generalization dynamics of simple ANNs.

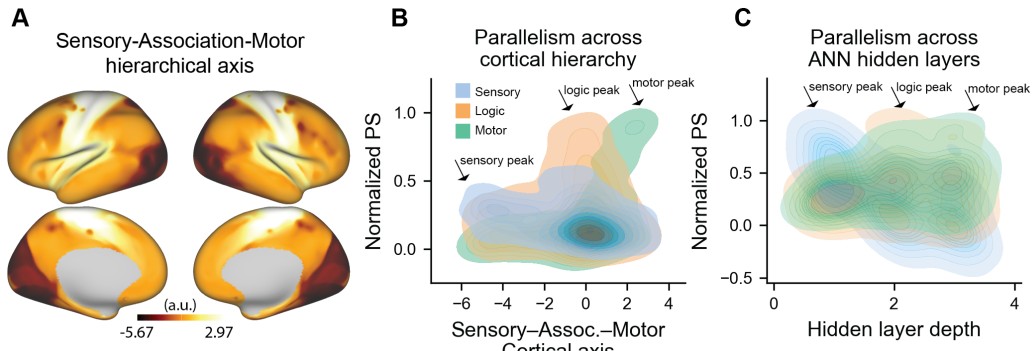

Supplementary Figure 5: Convergent hierarchical organization of abstract representations in humans and ANNs. a) We compared the topographic differences in PS across cortex with a well-known sensory-to-motor hierarchical gradient identified during resting-state fMRI [32]. b) We compared how content-specific abstractions (PS) differed across this hierarchical gradient, finding that sensory rule abstractions were highest in the lower part of the gradient (sensory systems), logic rule abstractions were highest across association cortex, and motor rule abstractions were highest across motor cortex. c) For an analogous analysis, we plotted how domain-specific abstraction differed across different hidden layer depths in the pretrained ANN with 3 hidden layers. We identified similar patterns of parallelism across the 3 hidden layers for each of the rule domains.

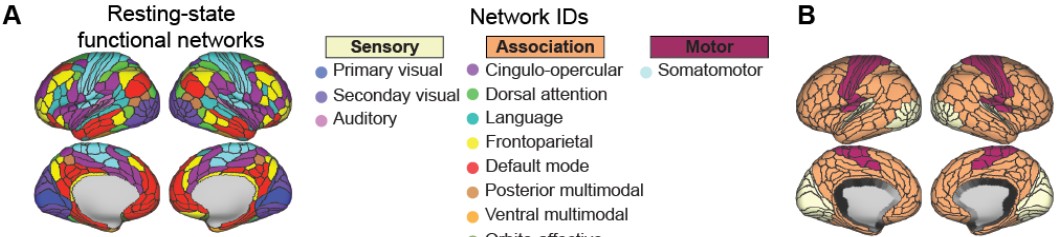

Supplementary Figure 6: a) A network partition of cortical parcels using resting-state fMRI [24]. b) To match the number of layers in the ANN, we created three discretized systems based on the functional network partition – sensory, association, and motor – that followed the sensory-to-motor hierarchy (Fig. 8) [32, 21]

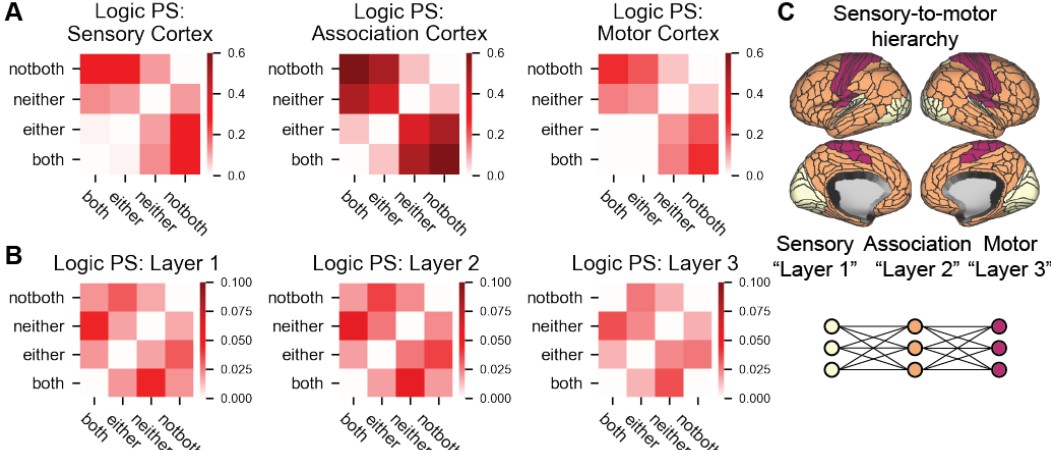

Supplementary Figure 7: A pairwise comparison of PS for all possible dichotomies in the Logic rule condition. a) We computed the PS for every pairwise dichotomy in the Logic rule domain, and computed the average PS across brain regions within each cortical system (Sensory, Association, and Motor cortex). The average PS for each cortical system was computed by averaging the PS across all regions within that cortical system for every dichotomy. b) To compare how the pairwise dichotomies matched in pretrained ANNs (Combined pretrained only; no training on full C-PRO trials), we computed the PS for all dichotomies in each layer. In this particular experiment we used an ANN with three hidden layers to compare with Sensory, Association, and Motor cortical systems. c) The sensory, association, and motor cortical systems can be analogized to hidden layer depths in the ANN.

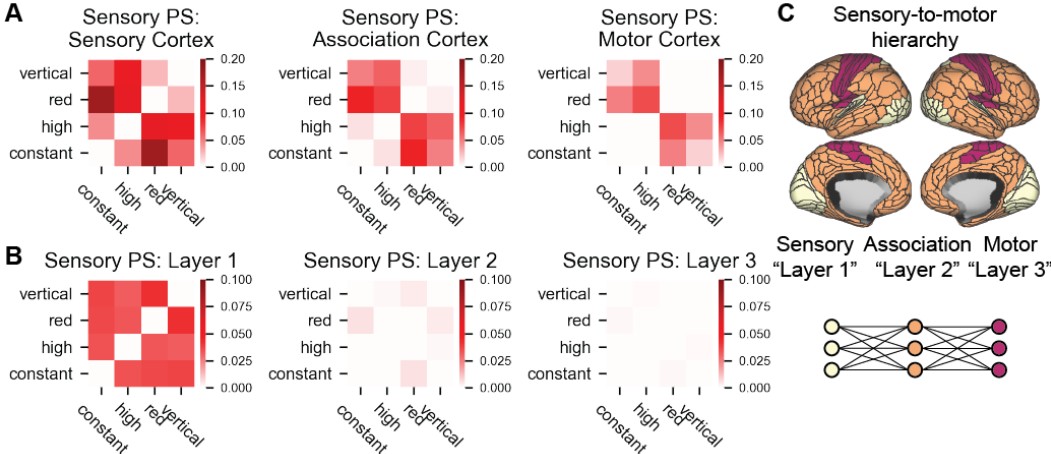

Supplementary Figure 8: A pairwise comparison of PS for all possible dichotomies in the Sensory rule condition. a) We computed the PS for every pairwise dichotomy in the Sensory rule domain, and computed the average PS across brain regions within each cortical system (Sensory, Association, and Motor cortex). The average PS for each cortical system was computed by averaging the PS across all regions within that cortical system for every dichotomy. b) To compare how the pairwise dichotomies matched in pretrained ANNs (Combined pretrained only; no training on full C-PRO trials), we computed the PS for all dichotomies in each layer. In this particular experiment we used an ANN with three hidden layers to compare with Sensory, Association, and Motor cortical systems. c) The sensory, association, and motor cortical systems can be analogized to hidden layer depths in the ANN.

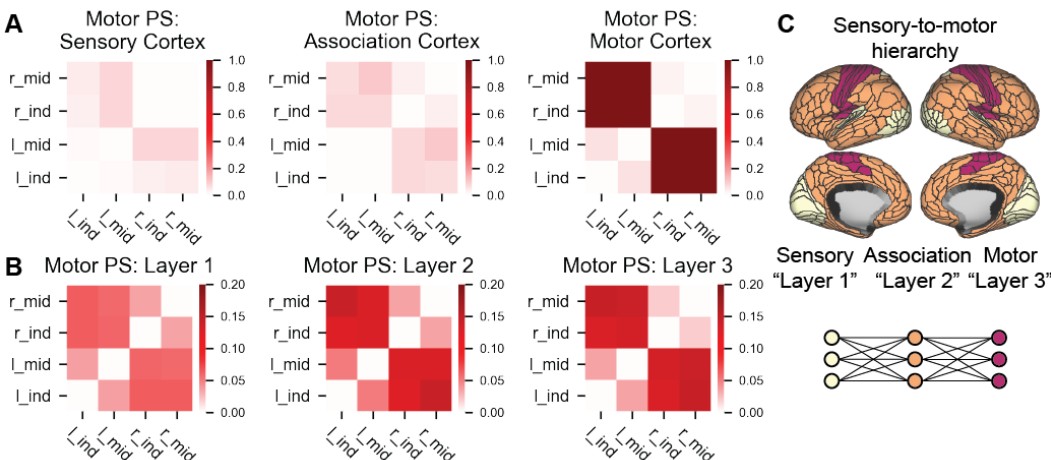

Supplementary Figure 9: A pairwise comparison of PS for all possible dichotomies in the Motor rule condition. a) We computed the PS for every pairwise dichotomy in the Motor rule domain, and computed the average PS across brain regions within each cortical system (Sensory, Association, and Motor cortex). The average PS for each cortical system was computed by averaging the PS across all regions within that cortical system for every dichotomy. b) To compare how the pairwise dichotomies matched in pretrained ANNs (Combined pretrained only; no training on full C-PRO trials), we computed the PS for all dichotomies in each layer. In this particular experiment we used an ANN with three hidden layers to compare with Sensory, Association, and Motor cortical systems. c) The sensory, association, and motor cortical systems can be analogized to hidden layer depths in the ANN.