# OpenReview forum: "Compositional generalization through abstract representations in human and artificial neural networks"
_NeurIPS.cc/2022/Conference — NeurIPS 2022 Accept_

### Official Review · Reviewer_CF9z · 2022-06-17

**Rating:** 7
**Confidence:** 3
**Soundness:** 3 good
**Presentation:** 4 excellent
**Contribution:** 3 good

**Summary:**

This work investigates compositional generalization in humans (using fMRI data) and ANNs. The working hypothesis of the paper is that parallel abstract representations - consistent variable representations across different contexts - support compositional generalization. The Parallelism Score, a measure from neuroimaging research, is used to measure the consistency of representations in humans and ANNs. In their experiments, the authors use the C-PRO paradigm to construct a 64-context compositional task for humans and ANNs. The paradigm enables systematically varying conditions to test generalization abilities. In their experiments, they expose humans and ANNs to limited contexts and test them in novel contexts. First they confirm that humans perform well on these tasks, but drop slightly in novel contexts. They observe parallel abstract representations in fMRI data, supporting their hypothesis that this enables generalization.  Based on their finding that for humans, increased exposure to a rule improves performance in novel contexts, they design pretraining tasks to improve generalization in ANNs. This includes 1-rule and 2-rule simplifications of the 3-rule tasks of C-PRO, enabling the ANN to learn prior representations for sub-rule variables. Without pretraining, ANNs struggle in novel contexts. Pretraining significantly improves performance in novel contexts, enables zero-shot performance, and more sample efficient learning. Pretraining enables compositional generalization in ANNs similar to humans and similarly increases parallel abstract representations. The authors also report similarities in the hierarchical Parallel Scores between human fMRI data and  ANNs.

**Questions:**

1. What is your response to weakness (1)? Do you think the relatively small increase in Figure 5A undermines your hypothesis that parallel abstract representations are important for compositional generalization?

2. Please elaborate on how you arrived at your pretraining method from your finding that in humans "increased exposure to specific rules improved performance on subsequent novel contexts". I understand the finding, and the pretraining method, but to me the link isn't obvious. The pretraining method does not simply expose the ANN to more examples of contexts in which the rule appears (that would simply be showing the ANN more 3-rule examples containing that rule). The pretraining method exposes the ANN to simplified contexts (1-rule or 2-rule examples), where it can focus on learning representations. How do you motivate the link?

**Limitations:**

The authors dedicate an entire section to discussing the limitations of their work. This is valuable and insightful.

**Strengths And Weaknesses:**

I think this is an excellent paper. The authors apply ideas from neuroscience research to the growing field of study that is compositional generalization in ANNs. My only caveat to confidently recommending acceptance is that I am not qualified to assess the validity of the neuroscientific claims made in the paper. But it seems that the authors have followed a solid methodology throughout and do not overstate any claims, so I would feel comfortable still recommending acceptance. Furthermore, the findings of the paper related to ANNs are quite interesting and novel on their own.

**Strengths**

1. The ideas in this paper are novel and valuable in the study of compositional generalization in ANNs. I am not aware of previous work applying the idea of parallel abstract representations to ANNs*, and this seems useful for future research. Furthermore, the methodology followed by the authors to perform analogous experiments in neuroimaging and ANNs is unique and well thought out. A human-centric benchmark for compositional generalization at such a scale (64 contexts) is certainly valuable.

2. For the most part, the claims made by the authors are supported and well motivated throughout the paper. Their pretraining method boosts generalization significantly and should be further developed in different deep learning contexts. I would have expected the PS measures on ANN hidden layers to be higher than those reported in Figure 5 A, but the zero-shot performance and sample efficiency makes a strong case that these pretraining methods are effective.

3. The paper is excellently written and well structured. The ideas are presented clearly, without overstating any claims.

\*  The closest idea to this that I have read of is the concept of “localism” from the paper by Hupkes et al. titled Compositionality Decomposed: How do Neural Networks Generalise? (2020). It’s not the same idea, but is similar in that it expects compositional models to build consistent variable representations independent of context.

**Weaknesses**

1. There is one experiment where the results are not as convincing as one would wish. In Figure 5A the PS measure increases only slightly after pretraining. This shows that pretraining does in fact encourage parallel abstract representations, but it does not necessarily show that this is absolutely crucial to compositional generalization. The PS measures increase slightly, but the generalization ability increases significantly, so perhaps some other factor enabled by pretraining is enabling generalization. Since the working hypothesis of the paper is the link between parallel abstract representations and compositional generalization, it might be worth discussing this result more.

2. The paper performs its ANN experiments with a simple 2-layer feedforward neural network. I understand that this isolates the effect of pretraining for the experiments, but in the process it ignores the neural network architectures that are currently state-of-the-art in most domains i.e. Transformers, CNNs, and LSTMs. Most existing studies on compositional generalization in ANNs have focused on these more advanced architectures and if this paper had done so it would have made a stronger case for the proposed methods and methodologies. As such, the findings of this paper are not directly comparable to leading edge research on the generalization capabilities of ANNs.

---

> ### Author Response · Authors · 2022-08-02
> **Response to Reviewer CF9z (1/1)**
>
> We would like to thank the Reviewer for the detailed discussion and encouraging comments. Below, we address each question asked by the Reviewer.
>
> **Question**: *What is your response to weakness (1)? Do you think the relatively small increase in Figure 5A undermines your hypothesis that parallel abstract representations are important for compositional generalization?*
>
> **Response**: The Reviewer raises an interesting point that we are happy to clarify. PS is obtained by averaging quantities based on pairwise coding vectors. Thus, it is expected to correlate on average with generalization, but it is not suited to capturing it in fine detail. For example, coding vectors can have non-coding or non-abstract dimensions (e.g., non-coding units within an ANN layer), which may mask the overall magnitude of PS. These non-abstract dimensions would introduce noise into the PS estimate, since they are not linearly additive. However, the existence of a few (or a subset) of abstract dimensions within an ANN layer or brain region might be sufficient for good generalization. These intuitions likely account for the discrepancies identified in Figure 5A and 5B.
>
> In addition, we acknowledge that parallel abstract representations present one potential mechanism that supports compositional generalization. However, this is not exclusive; there could be other potential mechanisms, such as program synthesis methods and neurosymbolic computing, which we discuss in the Introduction. Nevertheless, abstract representations – as captured with the PS – possess other important properties that we think deserve further investigation. For example, PS is a generic metric that is readily applied to both artificial and biological neural systems. In contrast, it is unclear how the identification of neurosymbolic modules/functions might be identified in empirical neural data. In addition, the evidence that a simple procedure (i.e., simple pretraining procedures) can realize these representational properties suggests that adhoc architectural elements like neuro-symbolic modules are not always necessary for compositional generalization.
>
> We would be happy to elaborate on these points in the final revision of the paper, provided the additional space in the camera-ready format.
>
> **Question**: *Please elaborate on how you arrived at your pretraining method from your finding that in humans "increased exposure to specific rules improved performance on subsequent novel contexts". I understand the finding, and the pretraining method, but to me the link isn't obvious. The pretraining method does not simply expose the ANN to more examples of contexts in which the rule appears (that would simply be showing the ANN more 3-rule examples containing that rule). The pretraining method exposes the ANN to simplified contexts (1-rule or 2-rule examples), where it can focus on learning representations. How do you motivate the link?*
>
> **Response**: In our ANN experiments, we manipulated two different levels of prior rule knowledge. The first was pretraining (in particular, Primitives pretraining). The motivation behind this pretraining procedure is to mimic the prior knowledge adult humans have prior to performing a new experimental task (e.g., humans have semantic knowledge about certain logical relations such as “BOTH” or “EITHER”, or semantic knowledge about what the color “red” is). We note that this prior knowledge is distinct from the “increased exposure to specific rules [which] improved performance on subsequent novel contexts”. The results pertaining to these claims (i.e., the impact of rule exposure on subsequent novel contexts) are found in the Appendix in Fig. 9. However, the text results are briefly detailed in Section 3.9: “Pretraining leads to compositional generalization in ANNs comparable to humans”. In this ANN experiment, ANNs were first trained on four C-PRO contexts (which was an identical experimental protocol performed in the human experiment), and then evaluated on the remaining 60 contexts. We found that novel contexts that had greater rule overlap with the 4 trained contexts had greater performance than novel contexts with fewer rule overlap. To make this distinction clearer, we have added text in section 3.5 to clarify this distinction: “We evaluated the learning and generalization dynamics of ANNs with and without pretraining, after training ANNs on 4 of the full C-PRO contexts. This matched the human experiment, since humans were exposed to 4 ``practice'' contexts prior to performing the remaining 60 novel contexts.”
>
> To summarize, pretraining is motivated as a way of mimicking the prior semantic knowledge humans have. Exposure to specific rules and C-PRO contexts are parallel experiments performed in both humans (Fig. 1b) and ANNs (Appendix Fig. 9).

---

### Official Review · Reviewer_h8RA · 2022-06-22

**Rating:** 6
**Confidence:** 4
**Soundness:** 4 excellent
**Presentation:** 2 fair
**Contribution:** 3 good

**Summary:**

Building off prior work proposing the role of “parallel abstract representations”—linearly additive vector representations of task dimensions—in compositional generalization, the authors set out to evaluate this framework through an analysis of existing behavioral and fMRI data as well as through ANN experimentation. The authors find that human task generalizability patterns are consistent with this factorized task representation framework and that parallel abstract representations are present for sensory, logic, and motor rule types in cortical systems that correspond to previous decoding work. They then propose a pretraining paradigm for ANNs on either task primitives or on simpler compositions of task primitives, and find that this improves zero-shot generalization to a more complex permuted rules task as well as sample efficiency, while resulting in representations which mirror some aspects of the neural encoding geometry and cortical hierarchy.

**Questions:**

The authors note (once again only in the appendix) that pretraining was stopped once pretraining tasks achieved 99%. To confirm, >99% accuracy on each task, e.g. until the worst task achieves 99% accuracy? How many training samples / iterations did this take under each pretraining paradigm?

For the Vanilla-ANN comparison, how is that generated? Is it just randomly initialized? From what distribution? (Okay I found this in appendix; add more details to main text.)

Aside from pretraining familiarizing the networks with task context rules, the networks also become more familiar with and learn information about the sensory stimulus input units. Do you think that this confound ends up being relevant for the observed increased sample efficiency, both for the primitives pretraining and especially for the simple task pretraining, which I assume was probably pretrained for more iterations to get to 99% due to the increased task complexity?

It would be helpful to see a better breakdown, in general, for both brain and ANN experiments, about which specific rules were more consistently encoded, e.g. how did PS scores differ across rules within each rule domain? Did certain dimensions drive mean PS scores more than others?

Overall, this is an interesting study with great potential for publication, but the authors really need to be more explicit and detailed in their description of the ANN pretraining experiments, especially in the main text, so as to alleviate reader concerns on potential confounds.


**Limitations:**

The authors have addressed some limitations of this work. I agree that future work should attempt to replicate this using other task paradigms.

**Strengths And Weaknesses:**

This paper addresses the interesting problem of compositional generalization, and does well to benchmark against human data. The extension of the “parallel abstract representations” framework to a new higher dimensional task and the resulting fMRI analysis is interesting, namely that these types of linearly additive vector representations of task abstractions appear to emerge in an expected and interpretable way.

The move to computational models and assessment of representational geometry in that regime is well motivated, and the results are very interesting, but in many respects, the details are drastically under-defined in the main text. In some cases, the reader is left struggling to understand critical aspects of what the authors actually did, until moving to the supplementary materials. For example, the precise exact input structure to the ANN is left to speculation until finding supplementary figure 8. This type of information needs to be expanded considerably in the main text.

---

> ### Author Response · Authors · 2022-08-02
> **Response to Reviewer h8RA (1/2)**
>
> We would like to thank the Reviewer for their thoughtful comments and useful suggestions to improve the presentation of the manuscript. In the revised version of the paper, we made sure to clearly indicate in the main text whenever important details were expanded upon in the Appendix. We have also moved some of these important sections highlighted by the Reviewer into the main text. Moreover, if the space constraints of the camera-ready version allow, we will move additional sections from the Appendix to the main text. Below, we individually answer the questions asked by the Reviewer.
>
> **Question**: *The authors note (once again only in the appendix) that pretraining was stopped once pretraining tasks achieved 99%. To confirm, >99% accuracy on each task, e.g. until the worst task achieves 99% accuracy? How many training samples / iterations did this take under each pretraining paradigm?*
>
> **Response**: Firstly, we apologize for not expanding on further details in the main text. As mentioned, in an effort to compromise between clarity and the space constraints of the main text, we have now gone through the main text to include explicit references to the specific Appendix sections that contain critical details. For example, in section 3.4, which details the referenced pretraining procedure in ANNs, we have now added this additional sentence: “Each pretraining condition was trained until ANNs achieved 99% accuracy (see A.8 for details).” Depending on the space constraints of the camera-ready version of the paper, we will transfer some of these details from the Appendix to the main text.
>
> We have also made additional clarifications in Appendix A.8:
>
> “Primitives pretraining was always performed prior to Simple task pretraining, unless otherwise noted. Pretraining procedures were blocked together, such that all conditions within the Primitives pretraining paradigm (i.e., Logic, Sensory, and Motor primitives pretraining) were trained until all three tasks achieved 99% accuracy. Simple task pretraining was subsequently performed until both Logical-Sensory and Sensory-Motor tasks were performed at 99% accuracy.”
>
> To directly address the Reviewers concern: Yes, >99% accuracy was required on each pretraining condition. For the Primitives pretraining, this included the Logic, Sensory, and Motor primitives pretraining. For the Simple task pretraining, this included the Sensory-Motor and the Logical-Sensory tasks. The number of training samples for each pretraining paradigm is reported in Figure 10 and 11 (unfortunately relegated to the Appendix due to space constraints). In brief, Simple task pretraining required significantly more samples than Primitives pretraining to achieve >99% accuracy (see Figures 10b and 11b at the point where the x-axis=0). Figures 10 and 11 also report the generalization performance and sample efficiency as a function of incrementally learning C-PRO contexts for each pretraining condition.
>
> **Question**: *For the Vanilla-ANN comparison, how is that generated? Is it just randomly initialized? From what distribution? (Okay I found this in appendix; add more details to main text.)*
>
> **Response**: We have now included some additional details in the main text (ANN methods section 2.4), as well as an explicit reference to the Appendix which contains further details:
>
> “The primary ANN architecture had two hidden layers (128 units each) and an output layer that was comprised of four units that corresponded to each motor response (see Appendix section A.7 and Fig. 8 for additional details). Training used a cross-entropy loss function and the Adam optimizer \citep{kingma_adam_2017}. The ANN transformed the trial input vector into a 4-element response vector with the equation $ Y = f_{ReLU}(X_{h}W_{h}+b_{h}) $. Weights and biases were initialized from a uniform distribution $U(-\sqrt{1/k},\sqrt{1/k})$, where $k$ is the number of input features from the previous layer.”

---

> > ### Author Response · Authors · 2022-08-02
> > **Response to Reviewer h8RA (2/2)**
> >
> > **Question**: *Aside from pretraining familiarizing the networks with task context rules, the networks also become more familiar with and learn information about the sensory stimulus input units. Do you think that this confound ends up being relevant for the observed increased sample efficiency, both for the primitives pretraining and especially for the simple task pretraining, which I assume was probably pretrained for more iterations to get to 99% due to the increased task complexity?*
> >
> > **Response**: The Reviewer is correct in assuming that Simple task pretraining required more samples than Primitives pretraining (see Fig. 10b and 11b). However, we do not think that the familiarization of task context rules with sensory stimuli during pretraining is a confound. On the contrary, we think that this part of pretraining is essential, and was the primary motivation for the sensory Primitives pretraining. In general, humans perform experiments with existing knowledge about mapping rule semantics to environmental features, such as knowing what the color “red” is, or how to perceive a “vertical” versus “horizontal” line. Pretraining without learning the mappings between stimulus features and sensory (or semantic) rules would therefore be too impoverished. A similar argument can be made for motor Primitives pretraining, where the network needs to learn the relationship between how motor rules correspond to actual motor actions. The novelty the ANNs are exposed to during testing on the full C-PRO task is in combining different rule features together, and teaching the network 1) novel combinations (systematic compositionality); 2) longer combinations of rules (compositional productivity). The increased sample efficiency in the Simple task pretraining case is likely more to do with the complexity of the task, since it requires the recombination of two rule domains, producing a total combination of (e.g., 4 Sensory x 4 Motor = 16 possible two-rule contexts). In contrast, Primitives task training only has 4 possible conditions per rule domain, reducing the overall number of possible conditions.
> >
> > **Question**: *It would be helpful to see a better breakdown, in general, for both brain and ANN experiments, about which specific rules were more consistently encoded, e.g. how did PS scores differ across rules within each rule domain? Did certain dimensions drive mean PS scores more than others?*
> >
> > **Response**: For completeness, we have now included 3 new Figures: Figures 15, 16, and 17, which illustrate the comparison of pairwise Parallelism Scores for all rule dichotomies in both human fMRI data for sensory, association, and motor cortical systems, and each ANN hidden layer (using a 3-layer hidden network for comparison). However, since we had no a priori hypotheses about how rule encodings should behave across different rule dichotomies, we have left the interpretation of these maps to the reader. However, we point out that some similarities of rule encodings can be seen across fMRI and ANN activity. In particular, the increasing abstraction among motor rules within the same hand responses becomes stronger in human motor cortex as well as layer depth in the ANN (Figure 17). We have included reference to these new Figures in the main text (Section 3.7): “(In addition, see Fig. 15-17 for PS scores for all possible rule dichotomies in human fMRI data and ANN activations.)”
> >
> > And also Appendix (section A. 9): “In addition, Figures 15, 16, and 17 illustrate that pairwise PS scores for all pairs of rule dichotomies in both ANNs and human fMRI data.”

---

### Official Review · Reviewer_HvQw · 2022-07-08

**Rating:** 7
**Confidence:** 4
**Soundness:** 3 good
**Presentation:** 3 good
**Contribution:** 2 fair

**Summary:**

The paper presents an analysis of fMRI task data which seeks to uncover "abstract representations" in the brain. This same analysis is then applied to a simple artificial neural network.

The analysis is built around a "parallelism score" which measures displacement vectors between brain activations. Given three modalities, (sensory, logical, and motor), each modality is held fixed and the other two are varied. The parallelism score for the fixed modality is an average measure of how aligned the displacement vectors are, averaged across combinations of the other two modalities.

The authors find that the highest motor parallelism is found in the motor areas, the highest visual parallelism in the visual areas, and that the logical parallelism is distributed throughout the brain, but lies mostly in the temporal and frontal areas.

This same analysis is applied to an artificial neural network. It is found that pre-training the NN results in better parallelism scores and better zero-shot performance.

**Questions:**

- line 105: What is mean by "coding axis"? In the text that follows, I gather that it is a displacement vector associated with a given modality. Is this correct?
- line 120: In this case, what is meant by "neural activation space"? The space of all fMRI scans?
- line 124: What is the activation vector? An vector of fMRI voxel inensity?
- Figure 3 caption: This is the first place where the activation vector is mentioned. I would move this into section 2.3.
- Figure 3 caption: "The cosine angle between two linear decoders" --> Is this the ANN mentioned in line 135? That seems like a non-linear decoder. What does it mean to take the cosine angle between decoders? Does this refer to the classification boundary? Or the hidden state representation?
- line 135: What is the input to the model? Is it the intensity for all voxels?
- Section 2.3: The PS will be high if the coding directions are parallel, but it will also be high if the coding directions are identical. How do we know that the coding directions do not all lie in the same low dimensional space, i.e., as in Figure 2B. Importantly, it would be good to know, of the pairs $v_i$ and $v_j$ that have high cosine similarity, what is the $l_2$ distance?
- Section 3.3: What is the input to these ANNs? A text description of the task?
- Section 3.4: What does it mean to measure the PS of an ANN? What is the activation vector for the ANN?
- line 217: Is there only one training example per context?
- line 220: "unlike humans (see Fig. 1b), ANNs with no prior knowledge cannot compositionally generalize." I understand the claim being made, but it should be noted that an adult human is not exactly without "prior knowledge"
- Figure 6: In 6B Brain systems are on the y-axis and rule domains are shown in groups. In 6C rule domains are shown on the x-axis and layers are shown in groups. I suggest putting layers on the y-axis in 6C to make the figures easier to compare.

## minor things
- line 171: missing Figure 7
- Section 3.5: missing Figure 9


**Limitations:**

Limitations are adequately addressed

**Strengths And Weaknesses:**

## Strengths
- Application of the parallelism score to neural networks is novel as far as I can tell
- Identifying the cognitive mechanism behind compositionality is an important problem
- Analysis of this particular fMRI dataset is novel and shows the promise of this sort of analysis

## Weaknesses
- My main concerns are about the significance of the machine learning results. The paper claims that it introduces a new pre-training scheme. But the model architecture being considered is a two layer feed forward network and the pre-training is similar to schemes that have already been proposed, e.g., gSCAN [1]. The simplicity of the architecture and setting make it hard to see whether the results about pre-training would still hold in general.
- The particular task setting also makes the comparison between human behavior and ANNs hard to interpret. The paper claims that their trained ANN shows a human-like response pattern to tasks. That is, the lowest level of the ANN is activated for sensory modalities, the middle layer for logic modalities, and the last layer for motor modalities. But this cannot be used to support the claim that the ANN has converged on a human-like hierarchy of representations, because there is no analogous ordered layering in the brain. Rather, the human brain results show that the motor areas respond to the motor modalities, the logic areas to the logic modalities, and the sensory areas to the sensory modalities.
- Some clarity in description would be helpful. In section 3.3, I am not sure what the model inputs and outputs are supposed to be.
- The stories about the brain and about the ANN don't seem to interact much. This paper presents some findings about the human brain and some findings about pre-training ANNs, but beyond section 3.7, the two stories do not seem very related.
- The paper talks about identifying abstractions, but the only abstractions that can be discovered using the parallelism score are ones in which the representations have a linear relationship.

## Bibliography
[1] Ruis, Laura et al. “A Benchmark for Systematic Generalization in Grounded Language Understanding.” ArXiv abs/2003.05161 (2020)

---

> ### Author Response · Authors · 2022-08-02
> **Response to Reviewer HvQw (1/4)**
>
> We thank the Reviewer for the thorough assessment of our manuscript, and for bringing our attention to the Ruis et al. paper, which we now reference in the revised manuscript. We first broadly address the weaknesses the Reviewer raised prior to responding point-by-point to each question.
>
> *Regarding the novelty and significance of the machine learning results.* The goal of the present study was to investigate the role of abstract representations for compositional generalization, and in parallel, characterize the topography of abstract representations in human fMRI data during the same task. We believed the best possible model (with fewest assumptions) to address this question was a simple connectionist model. This ensured that the representations that would emerge were not due to architectural choices that are associated with state-of-the-art models, such as the GECA-enhanced model reported in the gSCAN paper by Ruis and colleagues (now referenced in the revised manuscript). We agree that other machine learning models and studies achieve undoubtedly higher compositional generalization benchmarks than those demonstrated here. Nevertheless, we thought it would be important to 1) show that the representations learned by classic connectionist models can exhibit properties of compositional generalization (namely systematic and some productive generalization; Hupkes et al., 2020); 2) propose vector representations of rule abstractions that are exhibited both in connectionist models and the human brain. This presents a baseline from which future studies can explore the degrees of abstraction in more sophisticated models. For example, it would be interesting to explore if the prevalence of abstract representations in more sophisticated models corresponds to improved levels of compositional generalization.
>
> *Regarding the difficulty in comparing human and ANN data, the lack of analogous ordered layers in the brain (to ANNs), and the lack of interaction between the brain and ANN data.* While there is no explicit one-to-one correspondence between a simple feedforward ANN with the complexity of the human brain, we sought to use a hierarchical framework that would make comparison of ANNs with our human brain data possible. Conceptually, this is similar to how Yamins and colleagues compared representations in convolutional feedforward networks with the hierarchical organization of the primate ventral visual stream using neural spiking data (Yamins et al., 2014). Here we extend this analytic framework to the entire human cortical hierarchy. In the human neuroimaging literature, there are a number of studies that have worked to characterize the hierarchical organization (often called gradient organization) of the brain (for example, see Margulies et al., 2016; for a review, see Huntenburg et al. 2018). In particular, this gradient organization has revealed a hierarchy of functional specialization – from sensory cortex, to association cortex, to motor cortex – that is largely consistent with the flow of information during experimental tasks (e.g., subjects take in sensory information to produce a motor response and action). We used this sensory-association-motor gradient map directly from a prior study (Margulies et al., 2016), and used it to identify analogous layers in ANNs (e.g., analogize layer 1 to sensory cortex, layer 2 to association cortex, and layer 3 to motor cortex; see Figures 6, 13, and 14). While we recognize that this comparison is an oversimplification of hierarchical processing in the brain, it provides a tractable way to directly compare hierarchical representations found in ANNs with those identified in fMRI data.
>
> *Regarding the limitation that the Parallelism Score can only identify linear abstractions.* Indeed, the Reviewer is correct that this is a characteristic of the Parallelism Score. However, that is also by design; Bernardi and colleagues (of which the original metric was defined) proposed that vector representations that require non-linear readouts to be discriminated are not conducive to generalization (Bernardi et al. 2020). This argument relies on the classical idea of a bias-complexity trade-off according to which, if a downstream task needs a complex non-linear machine learning model to be learned, that will preclude generalization. It will nevertheless be interesting to explore in future studies the utility and feasibility of having nonlinear abstractions and their impact on compositional generalization.

---

> > ### Author Response · Authors · 2022-08-02
> > **Response to Reviewer HvQw (2/4)**
> >
> > Below we respond to the specific questions posed by the Reviewer.
> >
> > **Question**: *line 105: What is meant by "coding axis"? In the text that follows, I gather that it is a displacement vector associated with a given modality. Is this correct?*
> >
> > **Response**: Yes, this is correct. We have clarified that sentence: “Intuitively, PS identifies a coding axis (i.e., the parallel displacement vector) across task contexts that aids generalization.”
> >
> > **Question**: *line 120: In this case, what is meant by "neural activation space"? The space of all fMRI scans?*
> >
> > **Response**: The neural activation space refers to the set of voxel activations within a brain region used to calculate PS. (In prior studies, e.g., Bernardi et al. (2020), this activation space was the set of neurons in a functionally defined brain region (e.g., the prefrontal cortex).) We have clarified this sentence: “​​PS is defined as the cosine angle of the coding directions of the same rules in different contexts in the neural activation space (e.g., voxels or neurons within a brain region)”
> >
> > **Question**: *line 124: What is the activation vector? A vector of fMRI voxel intensity?*
> >
> > **Response**: Correct – we have clarified this in-text: “For each pair, we subtracted the fMRI voxel activation vectors associated with each context to obtain the vector that represented that coding direction (see Fig. 3a).”
> >
> > **Question**: *Figure 3 caption: This is the first place where the activation vector is mentioned. I would move this into section 2.3.*
> >
> > **Response**: We hope the ​in-text clarifications to the above questions help to clarify this caption.
> >
> > **Question**: *Figure 3 caption: "The cosine angle between two linear decoders" --> Is this the ANN mentioned in line 135? That seems like a non-linear decoder. What does it mean to take the cosine angle between decoders? Does this refer to the classification boundary? Or the hidden state representation?*
> >
> > **Response**: We apologize that this section was not clearer. The cosine angle between two linear decoders refers to two linear decoders trained to classify neural activation vectors of the same rule dichotomy (pair), but when the two rules are used in different contexts. For example, if BOTH and EITHER were to be classified, one linear decoder can be trained to classify this distinction in one set of contexts, and another linear decoder can be trained to classify this distinction in another set of contexts. So yes, this refers to the classification boundary between (for example) BOTH and EITHER. PS is equivalent to measuring the cosine angle of these two linear decoders. This method (or statistic) can be applied to either fMRI activation data or ANN data. These details are further expanded upon in the Appendix, but for brevity we have revised a portion of Figure 3’s caption to emphasize this metric can be applied to either fMRI or ANN activations.
> >
> > Figure 3 caption: “Intuitively, PS captures the geometry of the neural activation space (ANN or fMRI data) by measuring the cosine angle between two linear decoders trained to distinguish two rule conditions in different task contexts.”
> >
> > **Question**: *line 135: What is the input to the model? Is it the intensity for all voxels?*
> >
> > **Response**: The model was trained on the C-PRO task and was independent of fMRI data. We acknowledge the brevity of this section, so we have now revised line 136 to provide more explicit references to the Appendix and the Figure illustrating the model inputs/schematics in the Appendix to improve clarity: “The primary ANN architecture had two hidden layers (128 units each) and an output layer that was comprised of four units that corresponded to each motor response (see Appendix section A.7 and Fig. 8 for additional details).”

---

> > > ### Author Response · Authors · 2022-08-02
> > > **Response to Reviewer HvQw (3/4)**
> > >
> > > **Question**: *Section 2.3: The PS will be high if the coding directions are parallel, but it will also be high if the coding directions are identical. How do we know that the coding directions do not all lie in the same low dimensional space, i.e., as in Figure 2B. Importantly, it would be good to know, of the pairs $v_i$ and $v_j$ that have high cosine similarity, what is the $L_2$ distance?*
> > >
> > > **Response**: The Reviewer is correct: PS will be high if the coding directions are parallel, which can also happen if the representations are "clustered". For instance, if all the activation vectors in trials for logic rule BOTH cluster around the same point, and the activation vectors for the logic rule EITHER all cluster around another point (such that all coding directions with respect to the logic rule originate from and end at the same point), then PS will be high (Fig. 2B). Indeed, that is a feature of PS, which as originally developed (Bernardi et al., 2020), aimed to generalize the notion of clustering coefficient to characterize abstraction. One way to verify that a highly parallel representation is not merely a clustered representation is to perform a complementary decoding analysis, which aims to ensure representations are not just clustered for all rule pairs. If the representations are fully clustered, then they would not be discriminable/decodable. However, this is not the case, as we have reported a multi-class decoding analysis in Fig. 7 (in the Appendix).
> > >
> > > Regarding the $L_2$ distance between vector activations: PS ignores this, since it does not tell us whether the points that are being discriminated by a given coding direction are far or close (i.e., whether they can be decoded with high or low signal-to-noise ratio). A way to partially avoid this issue is to normalize the response vectors such that distances are in units of the noise variance. Although this method has the downside that it only works "on average" across coding vectors, as opposed to each coding vector individually, Bernardi et al. (2020) showed that empirically it works as well as more sophisticated methods like the Mahalonobis distance (which is essentially a multi-dimensional signal-to-noise ratio) in quantifying abstraction and generalization. Importantly, we applied this normalization prior to performing our decoding analyses in Fig. 7, such that distances are in units of the noise variance.
> > >
> > > **Question**: *Section 3.3: What is the input to these ANNs? A text description of the task?*
> > >
> > > **Response**: The input into the ANNs is a concatenation of one-hot vectors (i.e., concatenation of rule and stimulus one-hots). This is detailed in the Appendix (A.7) and Figure 8. See updated Line 183: “We constructed a simple feedforward ANN with two hidden layers (Appendix A.7; Fig. 8).”
> > >
> > > **Question**: *Section 3.4: What does it mean to measure the PS of an ANN? What is the activation vector for the ANN?*
> > >
> > > **Response**: The PS of an ANN is measured using the unit activations from the hidden layers. This was calculated separately for each layer. We have revised the main text to clarify this, while referencing the appropriate Appendix section (from Section 3.4): “PS was calculated for each rule domain using the ANN's hidden layer activations (see A.9).”

---

> > > > ### Author Response · Authors · 2022-08-02
> > > > **Response to Reviewer HvQw (4/4)**
> > > >
> > > > **Question**: *line 217: Is there only one training example per context?*
> > > >
> > > > **Response**: No, there are multiple training examples per context. Specifically, there are 256 possible unique training examples per C-PRO context. This is because there are 256 possible stimulus combinations. Every stimulus combination can be combined with a context to produce a unique context-stimulus grouping. Thus, in total, there are 256 unique stimulus combinations * 64 C-PRO contexts, producing a total of 16384 unique trials (i.e., context-stimulus combinations). For example, a stimulus sequence could be a red vertical bar paired simultaneously with a high pitch beep, followed by a blue horizontal bar with a low pitch beep. Depending on which sensory rule is included in the task context, irrelevant sensory information would be filtered out. We have now clarified this in the manuscript.
> > > >
> > > > Main text, Section 2.1: “One of 256 possible unique stimulus combinations could be presented with each task context. Visual dimensions included either horizontal or vertical bars with either blue or red coloring. Auditory dimensions included continuous (constant) or beeping high or low pitched tones... See Appendix A.2 for details.”
> > > >
> > > > We expanded on this in the Appendix, section A.2: “Visual stimuli included either horizontally or vertically oriented bars with either blue or red coloring. Simultaneously presented auditory stimuli included continuous (constant) or non-continuous (non-constant, i.e., ``beeping'') tones presented at high (3000Hz) or low (300Hz) frequencies. A given task context could be presented with 256 unique stimulus combinations. This is because a given task context was presented with two sequentially presented audiovisual stimuli, where each audiovisual stimulus varied in four dimensions: color (red/blue), orientation (vertical/horizontal), pitch (high/low), continuity (continuous/beeping). This led to $2^8=256$ possible stimulus combinations… ​​This meant that there were $256*64=16384$ unique trials (i.e., context-stimulus) combinations.”
> > > >
> > > > **Question**: *line 220: "unlike humans (see Fig. 1b), ANNs with no prior knowledge cannot compositionally generalize." I understand the claim being made, but it should be noted that an adult human is not exactly without "prior knowledge"*
> > > >
> > > > **Response**: We acknowledge the potential for confusion here, and so we have removed the clause “unlike humans”: “This suggested that ANNs with no prior knowledge cannot compositionally generalize.”
> > > >
> > > > **Question**: *​​Figure 6: In 6B Brain systems are on the y-axis and rule domains are shown in groups. In 6C rule domains are shown on the x-axis and layers are shown in groups. I suggest putting layers on the y-axis in 6C to make the figures easier to compare.*
> > > >
> > > > **Response**: We apologize for the error, and thank the Reviewer for pointing this out. In fact, the x-axis in panel B should be “Rule Domains”, making it consistent with panel C. In both panels B and C the colored legend within the graph reflects the Brain Systems and Layers, respectively. Brain Systems and Layers should be treated analogously. This error has been corrected.
> > > >
> > > > **Question**: *line 171: missing Figure 7”; “Section 3.5: missing Figure 9*
> > > >
> > > > **Response**: We apologize for the confusion. Figures 7-17 are included in the Appendix. In revision we will include both the main text and Appendix into the same PDF document.
> > > >
> > > > **References**
> > > >
> > > > ​​Bernardi, S., Benna, M.K., Rigotti, M., Munuera, J., Fusi, S., Salzman, C.D., 2020. The Geometry of Abstraction in the Hippocampus and Prefrontal Cortex. Cell. https://doi.org/10.1016/j.cell.2020.09.031
> > > >
> > > > Huntenburg, J.M., Bazin, P.-L., Margulies, D.S., 2018. Large-Scale Gradients in Human Cortical Organization. Trends in Cognitive Sciences 22, 21–31. https://doi.org/10.1016/j.tics.2017.11.002
> > > >
> > > > ​​Hupkes, D., Dankers, V., Mul, M., Bruni, E., 2020. Compositionality Decomposed: How do Neural Networks Generalise? Journal of Artificial Intelligence Research 67, 757–795. https://doi.org/10.1613/jair.1.11674
> > > >
> > > > Margulies, D.S., Ghosh, S.S., Goulas, A., Falkiewicz, M., Huntenburg, J.M., Langs, G., Bezgin, G., Eickhoff, S.B., Castellanos, F.X., Petrides, M., Jefferies, E., Smallwood, J., 2016. Situating the default-mode network along a principal gradient of macroscale cortical organization. PNAS 113, 12574–12579. https://doi.org/10.1073/pnas.1608282113
> > > >
> > > > Rigotti, M., Barak, O., Warden, M.R., Wang, X.-J., Daw, N.D., Miller, E.K., Fusi, S., 2013. The importance of mixed selectivity in complex cognitive tasks. Nature 497, 585–90. https://doi.org/10.1038/nature12160
> > > >
> > > > Yamins, D.L.K., Hong, H., Cadieu, C.F., Solomon, E.A., Seibert, D., DiCarlo, J.J., 2014. Performance-optimized hierarchical models predict neural responses in higher visual cortex. Proceedings of the National Academy of Sciences 111, 8619–8624. https://doi.org/10.1073/pnas.1403112111

---

> > > > > ### Comment · Reviewer_HvQw · 2022-08-10
> > > > > **Response to authors**
> > > > >
> > > > > I thank the authors for their thorough response.
> > > > >
> > > > > My three main concerns were (1) the correspondence that the authors were drawing between high/low level processing areas and layers of a network (2) the linear nature of the decoding and (3) the clustering of the learned rules.
> > > > >
> > > > > The responses give plausible reasons to think that (1) and (2) are given appropriate consideration for the scope of the paper and (3) has been handled by additional presented results.
> > > > >
> > > > > My score remains the same; I recommend for acceptance.

---

### Official Review · Reviewer_ETpS · 2022-07-11

**Rating:** 7
**Confidence:** 4
**Soundness:** 3 good
**Presentation:** 3 good
**Contribution:** 3 good

**Summary:**

Humans are good at generalizing previously acquired knowledge to find solutions to novel problems. In this way, they can learn a complex task by combining knowledge from different experiences instead of learning from scratch. This study suggests that neural networks can also afford this compositional generality by having task representations in a geometrical structure that allows a linear classifier to be used for generalization.


**Questions:**

Does the ANN learn different conditions in separate blocks for pretraining? How can the ANN learn a new condition while not modulating the weights learned from the pretraining of previous conditions (a.k.a. catastrophic forgetting)? Does the order of pretraining conditions matter in the accurate generalization? I think this part is the key to this paper but I can't find how this part was made.

**Limitations:**

Humans intuitively know which features of sensory stimuli can be combined and which features cannot be combined (e.g. red vertical stimulus can be shown but red blue stimuli cannot be presented). Therefore, they can generalize the representation of 'vertical' to 'blue' stimuli even only when they were trained with 'red' or when the colors of training stimuli are covered with their shape. However, the proposed ANN would not yet be able to overcome these kinds of biases. On the other hand, humans might have more experiences to answer to 'both' and 'either' than their negation, this would make it harder to respond to 'neither' or 'not both' conditions, while the ANN might not have  (or might not able to capture) these biases.

**Strengths And Weaknesses:**

Strength,
This work provides clear evidence that the representational structures (parallelism scores) are closely related to the ability for efficient cognitive control and generalization while the representations and cognitive control have been examined largely in separate domains. The effects of training facilitating fast learning can be found not only in human participants but also in the neural networks, which is interesting because this may not be easily explained by reinforcement learning but supports the idea that learning entails building abstract task structure representations.

Potential weaknesses,
It is not clear whether the neural representations were based on the data acquired at the time of the presentation of the conditioned cue, at the time of decision making (maybe ash the time of the second stimuli), or the mean activity across a mini block. If it is the former, this is before the stimuli presentations and decision making, which is not true in the ANN architecture (the ANN were made with the inputs of not only conditions but also stimuli). If it includes decision making, it may not be the rule but the activity associated with the decision making might be represented.

In the previous study (Bernardi 2020, et al.), the rule cannot be separated from the stimuli while the representational structure cannot be explained by the similarity between stimuli. That is, the structure was built based on associated rules for each of the visually independent stimuli. However, in this experiment, can some pairs of parallel representations also be explained by the (dis)similarity of input vectors? (e.g. the same pair of stimuli followed by the different action cues). How would the same condition be represented if they were paired with different stimuli pairs?

---

> ### Author Response · Authors · 2022-08-02
> **Response to Reviewer ETpS (1/2)**
>
> We thank the Reviewer for the thoughtful assessment of our manuscript.
>
> First, we address the weakness raised by the Reviewer: *The potential contamination of abstract rule representations by stimuli and/or motor responses in both the human fMRI and ANN activations.*
>
> We want to assure the Reviewer that abstract rule representations in both ANN and fMRI activations are not contaminated by stimuli/response activity.
>
> In the fMRI activations, we estimated the period of activity that is associated with the encoding period (i.e., presentation of the instructions). This diverges from the experimental design reported in Bernardi et al. (2020), where task context had to be directly inferred from the block of trials (since no explicit rule cues were presented). In the present experimental design (C-PRO task), instructions are presented for 3925ms, after which there is a variable delay (1570 - 6280ms), followed by the stimuli and response period (Fig. 1a). The fMRI activity was only estimated during the instruction period. This is reported in the Appendix, section A.6: “To extract task activations for each task block, we performed a beta series regression on every task miniblock [37]. Specifically, we fit an independent regressor to every encoding period (3925ms, 5 TRs), resulting in 128 task regressors in total. Fitting regressors on the encoding period was done primarily to isolate rule representations rather than the actual trial (stimulus-response) period.”
>
> In calculating the Parallelism Score (PS) from ANN activations, PS was calculated separately from the training procedure. Since we were only interested in the PS of rule representations, only input units associated with task rules were activated, while stimulus inputs were set to 0. We then computed a forward pass into the hidden units (again, with only rule units in the input space activated), and then computed PS based on these hidden activations. Thus, the hidden activations could not have been contaminated by stimulus inputs. We realize that we did not clearly specify this in the appropriate Appendix section (A.9) and thank the Reviewer for pointing that out. We have now added to Appendix A.9  the following clarification:
> “PS in ANNs was calculated separately from the training procedure. Since we were only interested in the PS of rule representations, only input units associated with the task rules were activated, while stimulus inputs were set to 0. This ensured that the hidden activations were not contaminated by stimulus-related activations when calculating PS in the hidden layers. Otherwise, PS in ANNs was calculated in a similar manner to the empirical fMRI data, where the spatial features (dimensions) were the units within a given hidden layer (like voxels within a brain parcel).”
>
> Next, we address the Questions raised by the Reviewer:
>
> **Question:** *Does the ANN learn different conditions in separate blocks in pretraining? How can the ANN learn a new condition while not modulating the weights learned from the pretraining of previous conditions (a.k.a. catastrophic forgetting)?*
>
> **Response:** Yes, the different conditions are presented in separate blocks in pretraining. However, we ensured that there was no catastrophic forgetting by repeatedly training on all condition blocks until satisfying a stopping criterion where all conditions (within each pretraining paradigm) achieved greater than 99.0% performance. This ensured that all pretraining conditions could be performed simultaneously with high accuracy. We have now revised the Appendix section describing the pretraining process (Section A.8) to include specific details of the pretraining process.
>
> “Pretraining procedures were blocked together, such that all conditions within the Primitives pretraining paradigm (i.e., Logic, Sensory, and Motor primitives pretraining) were trained until all three tasks achieved 99% accuracy. Simple task pretraining was subsequently performed until both Logical-Sensory and Sensory-Motor tasks were performed at 99% accuracy.”

---

> > ### Author Response · Authors · 2022-08-02
> > **Response to Reviewer ETpS (2/2)**
> >
> > **Question:** *Does the order of pretraining conditions matter in the accurate generalization?*
> >
> > **Response:** The Reviewer raises an interesting question. Prior evidence suggests that the order of pretraining is important for sample efficiency and generalization (in particular, ANNs are more sample efficient when transitioning from easier to more difficult tasks) (Saglietti et al., 2021). We sought to directly evaluate this in a simple follow-up experiment. We evaluated zero-shot performance and sample efficiency of “Combined” pretraining (i.e., an ANN model pretrained on the Primitives paradigm, then the Simple task paradigm), and then the reverse (i.e., “Reverse-combined”; Simple task pretraining, followed by Primitives pretraining). Interestingly, and consistent with prior studies (Saglietti et al., 2021), we found that the ordering significantly mattered for both generalization performance and sample efficiency. In other words, ANNs pretrained in the Reverse-combined condition could not generalize to the unseen C-PRO tasks, while the Combined condition could. Moreover, the Reverse-combined condition required more pretraining samples to achieve >99% accuracy on both the Simple task and Primitives pretraining paradigms, despite not affording any generalization performance on the C-PRO task. These new results are now reported and visualized in a new Figure 12, and mentioned in Section A.8:
> >
> > “Primitives pretraining was always performed prior to Simple task pretraining, except for Fig. 12, which investigated the effect of reversing the order of the pretraining tasks… This ordering is consistent with prior work, suggesting that ANNs are more sample efficient when transitioning from easier to more difficult tasks [41]. We also performed a simple control experiment demonstrating that when pretraining was reversed in the Combined condition (i.e., Simple task pretraining followed by Primitives pretraining), generalization performance was reduced to chance (Fig. 12). This suggests that the ordering of pretraining paradigms is crucial for generalization performance and sample efficiency, which future work should explore.”
> >
> > We hope that the revised manuscript satisfies the major questions and concerns from the Reviewer. Specifically, we hope that some of the new in-text revisions, which addressed that the estimation of PS in the present study cannot be confounded by stimulus-related or motor-related activity (as it might be in the Bernardi et al. 2020 study), satisfy the first main concern of the Reviewer. We also hope that the inclusion of the additional control experiment and the surrounding text (i.e. Fig. 12), clarifies potential confusions about the impact of pretraining protocols on C-PRO compositional generalization.
> >
> > **References**
> >
> > Saglietti, L., Mannelli, S.S., Saxe, A., 2021. An Analytical Theory of Curriculum Learning in Teacher-Student Networks. arXiv:2106.08068 [cond-mat, stat].
> >
> > Bernardi, S., Benna, M.K., Rigotti, M., Munuera, J., Fusi, S., Salzman, C.D., 2020. The Geometry of Abstraction in the Hippocampus and Prefrontal Cortex. Cell. https://doi.org/10.1016/j.cell.2020.09.031

---

> ### Comment · Reviewer_ETpS · 2022-08-08
> **Thank you for your answers. I rephrase my questions.**
>
> Thank you for kindly providing the answers to my questions.
> Let me rephrase my questions based on the answers.
>
> 1. I see that the rule representations are not driven by stimuli in both ANN and fMRI. My question also included how much the rule representations were driven by the input units of task context (e.g, the 4 one hot codes of sensory rules already can be represented in a generalizable structure with high PS in a 4 dimension space). If the PS score increases due to learning the task structure, The PS should be driven not only by the input units of task context but also increase during training. How does the PS score change according to the training steps? Does the score correlate with the performance accuracy?
>
> 2. Thank you for showing the simulation results while changing the training orders of simple and primitive tasks. That is very informative. My question was about how the ANN could learn different rules during primitives and simple task learning without interference from each other. It seems that each of the primitive rules has been learned in a block. If it is, I assume that the weights in hidden layers formed during the first block (e.g. learning both) would be changed while learning other rules in the later training block (e.g. red, left index, or right middle). If they are overwritten, though it reaches 90% accuracy during learning 'both' for block 1, they would not be able to reach the same level after learning other rules at block 4. That is, the ANN would perform better for the last learning rule compared to the earlier learning rule, which is the reason that I have asked if there is any bias based on the order of learning rules.
> This issue is discussed in some, for example, in Flesch et al 2018, https://www.pnas.org/doi/full/10.1073/pnas.1800755115.
> I must miss something. I would appreciate it if the authors point out what I miss here.
>
> 3. If I understood correctly, the PS was measured and tested between two vectors to test if a linear classifier can be generalized. To support the idea that the structure of rule representation was optimized for generalization I think it should be also shown the low PS score between orthogonal vectors. That has been tested in Bernardi et al for example by showing that a linear decoder can generalize only specific conditions but prevent over-generalization.
>
> 4. Is there any reason the input units of motor rules were constructed with four units instead of combinations of two units as the logic rules inputs were made? Can it be [1 for Left 0; for Right], and [1 for mid;  0 for index-finger]?
>
>
> Limitations: The compositional generalization would work only in a situation where the ANN should focus on only one sensory feature at a time. The compositional generalization across sensory rules is limited. It seems that the conceptual knowledge about 'red' can be used to understand the concept of 'blue' (~red). However, it is not quite like that. For example, the ANN may be able to make correct choices for 'both red and vertical' but it cannot make choices of 'both blue and vertical', which would not be the case for humans.

---

> > ### Author Response · Authors · 2022-08-09
> > **Response to additional questions, Reviewer ETpS**
> >
> > We thank the Reviewer for addressing our response. Below we address the additional questions.
> >
> > **Response to Q1**: This is an excellent question. The Reviewer is correct in that the rule input layers would already have high PS due to the way the inputs were represented – as one hot vectors. However, when calculating PS in the hidden layers, rule inputs were processed under the presence of noise. The network therefore needed to extract useful (i.e., abstract) task structure from noisy inputs. Moreover, it is important to consider that in the zero-shot performance case (i.e., Fig. 5), PS was measured in the presence of the full C-PRO task context (i.e., 3 rules), while the models were only pretrained using 1- (Primitives) or 2-rule (Simple) tasks.
> >
> > The Reviewer also suggested a great follow-up analysis – measure PS and C-PRO task generalization performance as a function of pretraining epochs/iterations. This would directly link the degree of PS and C-PRO generalization with pretraining. We have now performed this suggested analysis, finding a strong association across PS and generalization performance during pretraining iterations (r=0.88,p=10e-9). This is included a new Supplementary Figure (Fig. 18). If space allows, we will include this new figure into the main text in the camera-ready version, since we think it is a useful illustration. We thank the Reviewer for this thoughtful suggestion.
> >
> > **Response 2**: Indeed, the Reviewer is correct that each of the pretraining conditions (e.g., Primitives Logic rule training) is trained in separate blocks. However, the pretraining paradigm was not performed within a continual learning framework; the blocks were interleaved until **all** pretraining conditions achieved >99.0%.**Critically, this stopping criterion ensured that all pretraining conditions (e.g., Primitives Logic, Primitives Sensory, and Primitive Motor) could be simultaneously performed at a rate of >99.0% accuracy prior to exiting pretraining.**
> >
> > Prior work has suggested that multiple condition learning can occur particularly when condition blocks are interleaved, and when the learning rate is small enough (Fusi et al. 2007). We have now included citations to both Fusi et al. 2007 and Flesch et al. 2018.
> >
> > Appendix A.8: “Conditions within each pretraining protocol were interleaved \citep{fusi_neural_2007}, ensuring that catastrophic forgetting was not an issue, as is common in continual learning paradigms \citep{flesch_comparing_2018}.”
> >
> > **Response #3**: We think there may have been a misunderstanding, and apologize that our description of PS was not clearer. In the last revision, we tried to make the calculation of PS clearer. The cosine angle between two linear decoders refers to two linear decoders trained to classify neural activation vectors of the same rule dichotomy (pair), but when the two rules are used in different contexts. (Each task activation defines a point in the multivariate activation space, and the vectors are defined as the difference of these points.) For example, if BOTH and EITHER were to be classified, one linear decoder can be trained to classify this distinction in one set of contexts, and another linear decoder can be trained to classify this distinction in another set of contexts. We do not have any specific hypotheses about if the two vectors connecting cross-context activations were orthogonal, other than that the fact that these would not be cross-context generalizable. These details are further expanded upon in the Appendix.
> >
> > **Response #4**: The reason we did not reduce the motor rule inputs from four to two is that it would not have been possible to perform non-motor Primitives pretraining. This is because if 1 unit represented left or right hand with a 0 or a 1 respectively, and another unit represented index and/or middle finger with a 0 or a 1 respectively, by setting motor rule inputs to 0 during Logic rule Primitives pretraining, the network would infer that a the left index motor rule was presented. This would not appropriately allow for Logic or Sensory Primitives training without interference from motor rules.
> >
> > **Limitation:** *Compositional generalization was limited to situations in which ANNs focus only on one sensory feature at a time, and therefore is limited across multiple sensory dimensions.*
> >
> > **Response**: We agree that this is a limitation of the task, but do not think the inferences regarding abstraction (and PS) discussed here would be limited to single sensory features. The task in this study was to assess the mechanisms of compositional generalization across different rule domains. We think that these principles would similarly apply to compositions within the same sensory domain, when participants/ANNs are asked to attend to multiple sensory features (e.g., in a dataset such as COG or CLEVR). Nevertheless, this is an interesting empirical question that a future study can explore.

---

> ### Comment · Reviewer_ETpS · 2022-08-09
> **Update the scores according to the revised manuscript**
>
> Thank you for the authors addressing my questions and providing the results with additional analyses within a short time.
> Based on the author's response and revised manuscript, I change my score as below.
>
> Soundness: 2 -> 3
> Presentation: 2 -> 3
> Contribution: 3
> Rating: 5 -> 7
> Confidence: 2 -> 4

---

### Author Response · Authors · 2022-08-02
**Response to all Reviewers (summary)**

We are grateful to all Reviewers for their evaluation of our manuscript. We have now responded to each Reviewer, focusing on the Questions raised within each review, and where appropriate, addressing points raised as Weaknesses.

In addition, we have uploaded a revised manuscript (with the revised Appendix in one PDF) that addresses most of the Questions and Weaknesses. Most notably, we have performed several new analyses that address the importance of task ordering in the pretraining protocol (Appendix Figure 12), and the Parallelism Scores for every rule dichotomy (rather than just the average across all rules) in the ANN and fMRI data (Appendix Figures 15-17).

---

### Meta-Review · Area_Chair_bxcR · 2022-08-29

**Recommendation:** Accept
**Confidence:** Certain

**Metareview:**

The manuscript presents a story about compositionality that ties together neuroscience and models; with a focus on how compositionality enables generalization in a task performed by humans undergoing fMRI. Reviewers were largely happy with the manuscript and authors thoroughly addressed the questions that reviewers had.

The manuscript is suggestive. The experiment is in many ways limited, and it's not clear what conclusions one can draw at present about the design of models or of the brain. But, it is likely a significant audience at NeurIPS will be interested in this topic and it may spark followup work. This looks like the beginnings of an interesting line of work expanding the paradigm of mapping between brains and artificial neural networks.

**Award:**

No

---

### Decision · Program_Chairs · 2022-09-14

Accept